# Ruminant inner ear shape records 35 million years of neutral evolution

Bastien Mennecart [1] ✉, Ilya Dziomber [2,3], Manuela Aiglstorfer [4], Faysal Bibi [5], Daniel DeMiguel [6,7,8], Masaki Fujita[9], Mugino O. Kubo[10], Flavie Laurens[11], Jin Meng [12], Grégoire Métais[13], Bert Müller [14], María Ríos[15], Gertrud E. Rössner[16,17], Israel M. Sánchez [8], Georg Schulz [14,18], Shiqi Wang [19] & Loïc Costeur [1]

Extrinsic and intrinsic factors impact diversity. On deep-time scales, the extrinsic impact of climate and geology are crucial, but poorly understood. Here, we use the inner ear morphology of ruminant artiodactyls to test for a deep-time correlation between a low adaptive anatomical structure and both extrinsic and intrinsic variables. We apply geometric morphometric analyses in a phylogenetic frame to X-ray computed tomographic data from 191 ruminant species. Contrasting results across ruminant clades show that neutral evolutionary processes over time may strongly influence the evolution of inner ear morphology. Extant, ecologically diversified clades increase their evolutionary rate with decreasing Cenozoic global temperatures. Evolutionary rate peaks with the colonization of new continents. Simultaneously, ecologically restricted clades show declining or unchanged rates. These results suggest that both climate and paleogeography produced heterogeneous environments, which likely facilitated Cervidae and Bovidae diversification and exemplifies the effect of extrinsic and intrinsic factors on evolution in ruminants.

The Earth has undergone a general declining trend in global temperatures for the last 45 million years[1]. This has led to the establishment of permanent polar ice caps, which have strengthened the climatic belts leading to widespread continental seasonality and aridification. Climate is a major factor influencing distribution and diversity of life on the planet and it plays a cardinal role in speciation and extinction, best recognized on macro-evolutionary or geological time scales[2-4]. Moreover, climate variability, in particular in periods of declining global temperatures, is known to induce environmental heterogeneity and is hereby a strong selection factor. Changing climatic conditions and related tectonic activity (i.e., changes in local/regional to continental scale paleogeography) involve environmental shifts, modification in resource availability, and habitat fragmentation, paving paths for innovations to arise[5] and offer potential for diversification[6]. An example of this influence can be observed throughout the last 5 million years of human evolution[7].

Taxonomic diversity and body mass are major parameters used for inferring evolutionary rates and diversification episodes in mammals in deep time (e.g., ref. [8],). However, these proxies have significant drawbacks that prevent them from being effective, especially for determining reliable evolutionary rates. Body mass is a parameter known to be strongly correlated with ecological factors. Extinct taxa are often represented by fragmentary remains only, with a generally discontinuous record leading to difficulties in tracing the entire evolutionary history of a clade. Indeed, teeth are mostly well-preserved and widely distributed in the fossil record and represent one of the only possibilities to establish correlation between diversification and extrinsic factors on large scales and for whole clades[9]. However, they are also directly linked to feeding habits, making them prone to ecological plasticity. Investigating the entire evolution of a clade requires focusing on a structure that evolves neutrally through time. The bony labyrinth is an anatomical structure that has received attention recently in evolutionary research and preliminarily proven to

accurately reflect the kinship (i.e., phylogeny) between species —and hence their evolution over time[10,11]. The bony labyrinth is preserved as a hollow structure inside the petrosal bone, which is the densest hard tissue of the mammalian skeleto-dental system after teeth. It thus preserves extraordinarily well in the fossil record. Since the advent of CT-scanning techniques, allowing an assembling of a virtual endocast, the petrosal bone and the bony labyrinth have helped to clarify various aspects of mammalian evolution. Examples are found in the early steps of mammalian evolution in the Mesozoic[12], the mammalian phylogeny reconstructed using the wealth of morphological characters they can yield[10,13,14], or the evolution and dispersal of modern humans from Africa[11]. In addition, inner ear functions for hearing and balance are critical in the interaction between and within animals and their environments. Yet, data show that bony labyrinth morphology may only partly depend on ecological and physiological parameters such as habitat type[15,16], or locomotion capabilities[17,18]. Predominantly, the morphological signal recorded in the bony labyrinth is strongly related to phylogeny[10,19] and is considered to reflect neutral evolution[20]. The low selection pressure on this morphological structure may be due to its important developmental constraints[10].The bony labyrinth may thus be an appropriate proxy to investigate deep-time neutral evolution in vertebrates.

Ruminants are one of the most diverse groups of large mammals living today. They have evolved under the climatic fluctuations of the last 45 Ma from tiny forms without cranial appendages to the modern group with a diversity in body sizes, locomotor capacities, and cranial appendages (i.e., pronghorns, ossicones, antlers, or horns). Ruminants are abundant in the fossil record and very well represented in paleontological and neontological museum's collections. These collections document their survival through the worldwide Eocene-Oligocene cooling and aridification event 34 million years ago and show how they successfully adapted to new habitats. Thereafter, they highly diversified during the Miocene Climatic Optimum ca. 18–15 Ma when cranial appendages first occurred in all modern groups[21]. They have distributed and diversified in South America at the onset of the ice age ca. 3 Ma during the Great American Biotic Interchange (GABI) and, finally, became closely linked to human evolution. Ruminants have thus not only accompanied and survived the major climatic and geological events of the Cenozoic, but highly benefited in species diversity. Through these radiations, they became one of the taxonomically richest components of the terrestrial vertebrate fauna. Yet, little is known about how they precisely reacted to environmental changes and how their diversification and radiations were impacted by intrinsic and extrinsic factors. Due to intrinsic factors related to ecological optimum (e.g., metabolism), the evolutionary rates of the bony labyrinth morphology may differ from one clade to the other depending on Cenozoic climatic events (extrinsic factors), highlighting the current picture of the ruminant differences observed in their diversity and success. For all of these reasons, we use the ruminant bony labyrinth shape as a proxy to explore how the evolution of the different ruminant lineages have been impacted by intrinsic and extrinsic factors.

In this study, we investigate if the bony labyrinth is a good proxy to analyze the phylogeny and evolution of a whole mammalian clade, directly including fossil data into the analysis. Since bony labyrinth morphology may partly evolve neutrally and its size evolution may be more strongly related to ecological factors, a comparison between both parameters is likely to clarify different aspects of ruminant evolution. In addition, we test if bony labyrinth morphological evolutionary rates help understand the relationships between intrinsic and extrinsic factors shaping the evolutionary processes. Our results show that bony labyrinth shape morphology compares well to molecular-based phylogenetic hypotheses. Bony labyrinth morphology and size display different evolutionary scenarios. Both the increase of maximum size, according to the Depéret-Cope's rule, and an increase of the evolutionary rate of size, are demonstrated in the younger taxa of almost all clades. The morphological evolutionary rates of the bony labyrinth through time vary depending on the considered clade. Bony labyrinth morphological evolutionary rates clearly react to intrinsic factors responding to their respective ecological optimum combined with environmental and geographical conditions.

## Results/Discussion

### Principal component analysis

The PCA polymorphospace possesses a strong phylogenetic signal (permutation test $p < 0.001$). The average amount of shape change along the branches of the phylogenetic tree is indeed very small, in comparison to the null hypothesis of no phylogenetic signal. The dataset displays similar covariances among species to the covariances expected under Brownian motion (Pagel's λ 0.83, $p < 0.001$). However, the variance is mostly within the clade, not between clades (Bloomberg's K 0.36).

21.99% of bony labyrinth shape disparity is present along the PC1 axis (Supplementary Data 1). The relative size of the anterior and posterior semicircular canals in comparison to that of the cochlea is smaller in the most negative values of PC1 in comparison to the most positive ones. The anterior and posterior semicircular canals are more rounded in the most negative PC1 values than the positive ones where they are dorsally extended. The lateral semicircular canal in the most negative values of PC1 branches dorsally above the posterior ampulla, in the vestibule between the latter and the base of the common crus (Supplementary Data 1). In the most positive values of PC1, the lateral semicircular canal directly branches in the posterior ampulla. In the most negative values, the vestibular aqueduct runs parallel to the midline of the common crus along its course and extends above the end of the common crus dorsally. In the most positive values, the vestibular aqueduct is curved and shorter than the common crus (Supplementary Data 1). The PC2 axis represents 10.60% of the data variation. The bony labyrinth located in the most positive values of PC2 are displaying a bigger cochlea than in the most negative PC2 values, especially considering the first turn of the cochlea (Supplementary Data 1). In the most negative values of PC2, the ovoid lateral semicircular canal directly branches in the posterior ampulla. In the most positive values of PC2, the rounded lateral semicircular canal branches anteriorly to the posterior ampulla in the vestibule. The fenestra vestibuli is narrower in the most negative values of PC2 than in the positive ones. The vestibular aqueduct is extremely short and curved in the most negative values of PC2, while it is straight and longer than the common crus in the most positive values.

The most negative values along PC1 are occupied by tragulids, while members of Pecora have higher values (Supplementary Data 1). No clear trend can be observed within Pecora along the PC1 axis. Along the PC2 axis, the negative values are mostly occupied by bovids within the Pecora. The positive values of PC2 are occupied by all members of Pecora and Stem Ruminantia without clear distinction. The Bovidae represents the largest morphospace within the dataset, having the maximum and minimum PC1 and PC2 values (Supplementary Data 1). When PC1 vs PC2 scores are further inspected through time, we find that the Early-Middle Miocene *Dorcatherium* specimens have the highest values along the PC1 axis within the Tragulidae, marking a transition between the Stem Ruminantia and the modern Tragulidae (Supplementary Fig. 1). Interestingly, until the transition between the Middle and Late Miocene, Pecora display a similar shape disparity along the PC1 and the PC2 axes of Stem Ruminantia (Supplementary Fig. 1). The morphological increase in disparity of the Bovidae also took place during this period and diverges taxonomically in the fossil record.

## Between-group PCA

The overall classification accuracy when considering the bg-PCA is 58.5 % (Supplementary Data). Stem Ruminantia, Tragulidae, Antilocapridae, Dromomerycidae, and Cervidae are above 60% of classification accuracy with the Tragulidae culminating at 100%. In contrast, Giraffidae, Moschidae, and Stem Pecora are below 50% of classification accuracy with Moschidae having the lowest cross-validated classification result (36.4%). Stem Pecora are largely reattributed as stem ruminants (27.3%), while Giraffidae are reattributed as Dromomerycidae (12.5%).

The bg-PC1 axis represents 70.92% of the variance (Supplementary Data 1). A clear distinction between the Tragulidae and all the remaining ruminant groups can be observed. The Tragulidae bony labyrinth differs from that of the other ruminants by having a longer cochlea, a longer common crus than the vestibular aqueduct, and a very oblique lateral semicircular canal branching in the vestibule between the posterior ampulla and the base of the common crus. Along the bg-PC2 axis (12.85%), the negative values are mostly occupied by Bovidae, while Stem Ruminantia and Stem Pecora mostly occupy the highest values along bg-PC2 (Supplementary Data 1). All other ruminant groups overlap along bg-PC2 scores. The stem ruminants possess a long cochlea, a large fenestra vestibuli, the base of the vestibular aqueduct is anterior to the axis of the common crus, and the posterior semicircular canal is remarkably higher than the anterior semicircular canal. In the Stem Pecora, the cochlea is shorter, the posterior and anterior semicircular canals are of similar height, and the lateral semicircular canal branches high in the posterior ampulla. The Antilocapridae, Giraffidae, and Cervidae are relatively close morphologically to the Stem Pecora. Members of the Bovidae are generally characterized by a short and slightly oblique vestibular aqueduct and a very oblique lateral semicircular canal that branches, similarly to members of the Moschidae, in the vestibule anterior to the posterior ampulla.

When observing bg-PC1 vs bg-PC2 through time, the Early-Middle Miocene *Dorcatherium* specimens have the lowest values along bg-PC1 axis within the Tragulidae, making a transition between the Stem Ruminantia and the modern Tragulidae (Supplementary Fig. 2). The Stem Pecora and the Stem Ruminantia display a relatively similar mean shape along the bg-PC1 and the bg-PC2 axes (Supplementary Fig. 2). A first shape shift occurs for the Crown Ruminantia during the Early Miocene. Considering the Bovidae, similarly to their disparity of shape (PCA), a shift in the characterizing shape of the family can be traced back to the transition between the Middle and Late Miocene.

## Canonical variates analysis

The overall classification accuracy when considering the CVA is 80.4 % (Supplementary Data 1). Stem Pecora, Tragulidae, Antilocapridae, Cervidae, and Bovidae are above 60% of classification accuracy with the Tragulidae culminating at 96.6%. Stem Ruminantia, Giraffidae, and Dromomerycidae, however, are below 50% of classification accuracy. Dromomerycidae, wich have a null cross-validated classification result, could equally be confused with Stem Pecora, Giraffidae, and Cervidae (33.3%). Stem Ruminantia can be confused with Stem Pecora (30.0%), while Giraffidae may be confused with Bovidae (37.5%).

The CV1 axis represents 45.2% of the total variance (Fig. 1). The Tragulidae are isolated among the highest CV1 values (15 to 20). All other ruminants are located under the CV1 value of 5. We can observe a trend towards the lower values in the repartition of the different clades starting from the Stem Ruminantia, Stem Pecora, Moschidae, Antilocapridae, Giraffidae, Cervidae, and Bovidae. This trend is even clearer considering CV1 vs CV2 through time (Fig. 1). Along the CV2 axis (16.9%), we can observe a similar trend in the repartition of the data. The position of the Tragulidae is just intermediate to the Cervidae and Bovidae. The Tragulidae bony labyrinth differs from that of the other ruminants by having a longer cochlea, a longer common crus than the vestibular aqueduct, and a very oblique lateral semicircular canal

branching in the vestibule between the posterior ampulla and the base of the common crus. Along the CV2 axis (16.9%), the negative values are mostly occupied by Bovidae, while Stem Ruminantia occupy the highest values (Supplementary Data 1). A trend can be observed with the Giraffomorpha and the Cervidae having the lower CV2 positive scores, the Moschidae and Antilocapridae having slightly higher CV2 scores, and finally the Stem Pecora having just lower CV2 scores than the Stem Ruminantia. The Stem Ruminantia and Stem Pecora possess an enlarged cochlea, longer in Stem Ruminantia than Stem Pecora, and a large rounded fenestra vestibule. The base of the very elongated vestibular aqueduct is anterior to the axis of the common crus, and the lateral semicircular canal branches high in the posterior ampulla. In Stem Ruminantia, the posterior semicircular canal is remarkably higher than the anterior semicircular canal, while in the Stem Pecora they are of similar height. The Antilocapridae, Giraffomorpha, and Cervidae are relatively close morphologically to the Stem Pecora. Nevertheless, Antilocapridae differ markedly from the other Crown Pecora by retaining a large rounded fenestra vestibule, while Giraffomorpha, Cervidae, Bovidae, and Moschidae possess an ovoid fenestra vestibule. Members of the Bovidae are generally characterized by a short and oblique vestibular aqueduct and a very oblique lateral semicircular canal that branches, similarly to members of the Moschidae, in the vestibule anterior to the posterior ampulla. Posterior and anterior semicircular canals are relatively of the same height in Bovidae, while the anterior one is higher than the posterior one in Moschidae. Their vestibular aqueduct is longer than in Bovidae.

## Inner ear shape reflects phylogeny

As expected according to previous analyses of the bony labyrinth in vertebrates[10,15,22–24], the impact of the phylogeny on the shape of the bony labyrinth is significant in ruminants. The bg-PCA shows that Pecora, except Bovidae younger than 10-9 Ma, have retained an overall bony labyrinth morphology similar to that of the most ancestral stem ruminants studied (Supplementary Fig. 2). The speciose extant bovids (more than 139 living species[25]) started their long-lasting diversification in the Late Miocene, expressed in taxonomy and reflected in disparity of their bony labyrinth morphology. They colonized very diverse environments and habitats, hereby exemplifying the largest

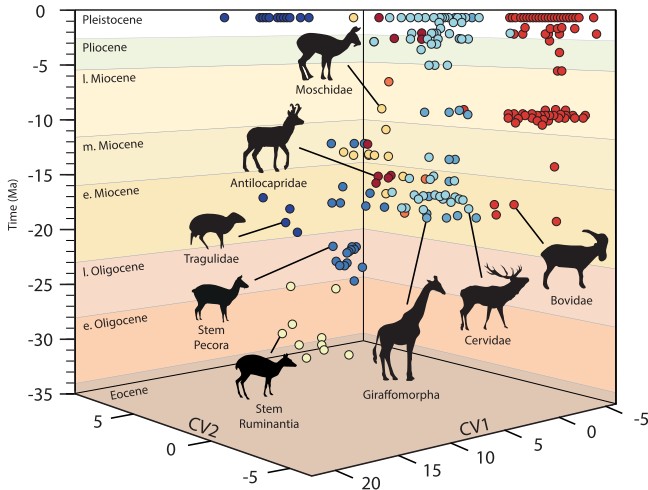

**Fig. 1 | Evolution of ruminant bony labyrinth morphology through time.** Shape changes are observed along CV axes (CV1 45.23% & CV2 16.84%). Groups are Stem Ruminantia (light yellow), Tragulidae (dark blue), Stem Pecora (light dark blue), Antilocapridae (dark red), Dromomerycidae (orange), Giraffomorpha (dark light blue), Cervidae (light blue), Moschidae (yellow), and Bovidae (red). Bivariate plot of CV1 vs CV2 and animated gif of the Fig. 2 can be found in Supplementary Data 1. Thickened circles are the bony labyrinth specimens from Supplementary Fig. 3. Silhouettes of the families modified from ref. 107.

ecological diversity among Ruminantia[26]. The bony labyrinth of tragulids (the only living non-pecoran ruminants) was morphologically derived early in its evolutionary history and has remained thus conserved thereafter[10] (Fig. 1). Maximizing the separation of the between-group means in relation to the variation within the groups' ratio according to specified selected clades, we clearly observe a shape continuum through time from the basalmost and oldest stem ruminants to different pecoran families (Fig. 1). The bony labyrinth morphologies characterizing the Pronghorn are the closest to those of the stem pecorans, while those of bovids are more derived, confirming the consensus reached by molecular phylogenetic hypotheses[27–29].

## Rates of evolution show radiation events

Ruminants went through several episodes of radiation in relation to environmental changes[10,21,28]. The evolutionary rates of the bony labyrinth morphology reveal different scenarios for the different clades of ruminants. The Early Miocene emerges as a period when evolutionary rates show accelerations at the base of crown pecoran clades, especially in bovids and giraffomorphs (Fig. 2). This has already been supposed based on molecular data[28]. The rapid morphological diversification at the base of these pecoran clades may represent the effect of an ecological opportunity[30] since only these pecoran clades entered Africa at this precise period, while few other ruminants are known prior to that time from Africa[31–33]. Other more significant evolutionary accelerations occur at terminal nodes in clades with recent radiative events (Supplementary Data 1). These include duikers (Cephalophini, of which 19 species arose in the last million years in the forests of central Africa[25,34]) or Odocoileini in cervids. These examples confirm that rates of speciation and morphological evolution are positively correlated on macroevolutionary time-scales[35]. For example, insular taxa are known to strongly diverge morphologically from their mainland ancestors (e.g., Köhler and Moyà-Solà[36]). Indeed, clades including specimens and species that are insular since the Pleistocene (e.g., *Rusa timorensis*, *Cervus astylodon*, and *Cervus nippon* from Japan) reveal acceleration of rates in the evolution of the bony labyrinth. Bovids and cervids diversify greatly from the Middle Miocene onwards and start colonizing a large range of biomes. As an example, many caprine bovids are typical mountain dwellers. They probably originated on the rising Middle Miocene Tibetan Plateau and diversified in response to this new ecological opportunity[29] highlighted by an increase in their evolutionary rate (Fig. 2). Warm climate open environment dwellers such as the very speciose antelope clades and their most recent common ancestors are recorded in the latest Middle to Late Miocene of Africa[37,38]. Other clades display significantly lower evolutionary rates throughout their history, such as the Tragulidae (Supplementary Data 1), which are arising in the Eocene[39,40]. Based on the bony labyrinth, this clade remains a less morphologically diversified group throughout the Cenozoic, reflected by its general body bauplan. This is probably related to a narrower ecological plasticity and adaptive disadvantage in digestive physiology displayed by a more restricted distribution across climate zones than other ruminants[25,41,42]. Moreover, the skull morphology of this lineage shows paedomorphosis[43], which may explain the observed significant decrease in evolutionary rate.

## Morphological evolution and the correlations to climate and geological events

Morphological traits, like hypsodonty or appendage characters, have been shown to correlate to environmental changes[21]. This is not systematically the case considering the morphological evolution of the ruminant bony labyrinth. The phenotypic evolution of the bony labyrinth morphology was tested against global temperature using the *EnvExp* model implemented in *RPANDA* (see Material and Methods and[8]). Colder periods, such as at the Eocene-Oligocene boundary, the late Oligocene glaciations, or the early Late Miocene to the present, all

see accelerating rates of morphological evolution of the general pecoran bony labyrinth shape (Supplementary Fig. 4). These cold pulses and periods are known to have triggered the development of more arid and more heterogeneous environments and habitats, especially at mid-latitudes[44–46]. Such a fragmentation increases the diversity of potential habitats, stimulating clades to diversify ecologically and taxonomically[47]. Comparing different ruminant clades offers a more complex picture. The evolutionary rates for Tragulidae are constantly low and not supported by the climatic model ($\beta = 0$). This result indicates that they may have been more stenopotent and unable to expand into more arid or colder environments. Low bony labyrinth evolutionary rates may here express a certain phenotypic stasis, which is otherwise known in groups qualified as "living fossils", where morphology and the ecological niche have barely changed over millions of years. In contrast to this, we find a positive correlation in giraffomorphs ($\beta > 0$). Giraffomorphs have a very low diversity today with only two living genera restricted to African intertropical savannas and limited to subtropical forests. Despite a strong increase in global species diversity within crown Giraffidae during the Middle and Late Miocene, their rates of evolution decline after this period (Fig. 3). This concurs with a decrease of global temperatures and the extinction of stem giraffomorphs (palaeomerycoids and non-giraffid giraffoids), which testifies to a drop in morphological disparity due to the shrinking of their habitats. Biological factors must be postulated here. The less efficient longer gestation periods in giraffes may have accelerated the disappearance of these large ruminants[25] within the context of global cooling. Moreover, diversification is known to slow down when a large part of a clade disappears[48], accentuating the effect of evolutionary rate decrease. Both cervids and bovids benefited from the global temperature decline related to permanent polar glaciations through adaptation to more heterogeneous habitats and seasonal climates, which is reflected in the evolutionary rates of the bony labyrinth ($\beta < 0$; Fig. 3). In particular, increasing aridification related to the global cooling in Africa from the early Late Miocene onwards led to habitat heterogeneity, thus promoting an ecological opportunity for taxa through adaptive radiation and, ultimately, to the diversification of floras and faunas[49]. However, global temperatures alone cannot explain the huge increase in the evolutionary rate found in Cervidae during the Plio-Pleistocene ($\Delta$ AIC < 4 without any effect of smoothing parameters; Supplementary Data 1). Indeed, the last 3 million years experienced an explosion of cervid species diversity, especially in South America. During this time at least 19 extant species evolved regionally and which currently represent a third of total cervid species diversity. The interplay between a new ecological opportunity for North American deer, entering South America during the Great American Biotic Interchange[50], and environmental heterogeneity fostered by the progressive strengthening of seasonality[51], coincides with the accelerating rates of the bony labyrinth morphological evolution in cervids and their increase in taxonomic and morphological diversity documented over this short period. Bovids do also show an increase of their bony labyrinth evolutionary rates but to a lesser extent than cervids despite a higher taxonomical diversity today. Extrinsic factors involving the colonization of South America and an adaptive radiation may have been preponderant for cervids.

North American pronghorns (Antilocapridae) also show a rapid acceleration of their evolutionary rates during the last 3 million years. However, in contrast to cervids and bovids, they remained ecologically poorly diversified and inhabited open habitats throughout their entire history[52]. This may explain the relatively low evolutionary rates of the clade during most of its history. The drastic decrease in species diversity (with only one species being recorded today) and a possible evolutionary bottleneck[53] may explain a derived bony labyrinth morphology for *Antilocapra* in comparison to those of fossil antilocaprids (Fig. 4G, H, e.g., shape of the vestibular and cochlear aqueducts). This may have driven the acceleration in rates of morphological evolution.

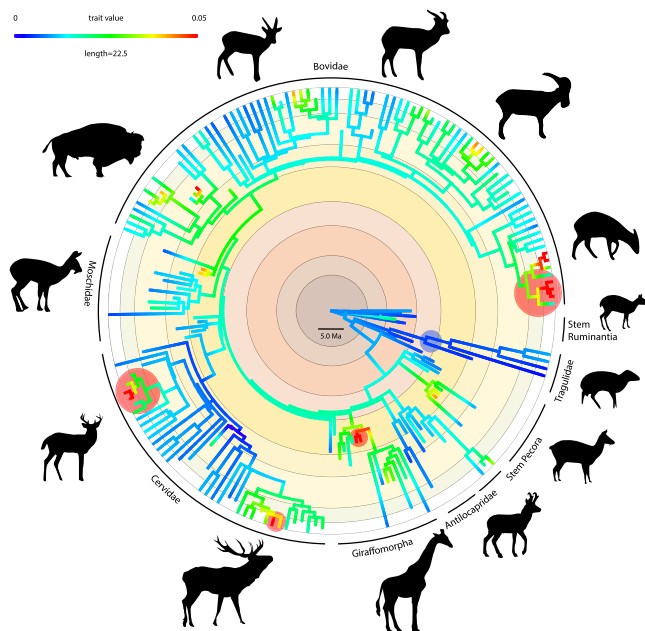

**Fig. 2 | Evolutionary rates of the ruminant bony labyrinth morphology based on PC scores through the phylogenetic tree.** Significant decrease in the evolutionary rates (blue circle) is observed within the tragulid evolution, while a significant increase of the evolutionary rates (red circle) is observed at the base of the Giraffomorpha during the Early Miocene and within the South American deer, insular cervini, and the duikers during the Pleistocene (Supplementary Data 1). Methodology and statistical results produced by the *R* packages *RRphylo*[91] and *Phytools*[88] are provided in Material and Methods and Supplementary Data 1. Silhouettes of the families modified from ref. [107]. Same color code as in Fig. 1 for ages.

Population size reduction, such as in bottleneck situations, is a major factor enhancing the influence of genetic drift and highlights the impact of neutral evolution on bony labyrinth morphology[20].

The evolution of bony labyrinth morphology in extant and extinct ruminants confirm that extrinsic factors (e.g., climate and tectonics), as well as intrinsic factors (biological traits combined with ecological opportunities) over long time scales have contributed in triggering the global morphological diversification and diversity of this group. The bony labyrinth shape can reflect functional requirements under strong selection pressure, best observed in extreme cases (e.g., dolphin[18]). However, a growing body of evidence shows that its shape is strongly correlated to phylogeny[10,19]. Neutral evolutionary processes seem to be driving its morphological evolution, visible at the microevolutionary level. This demonstrates that fine scale bony labyrinth shape differences in populations of closely related species are significant[22,54]. Our contrasting results on evolutionary rate shifts in various groups show striking similarities to molecular evolutionary rates. They confirm that bony labyrinth shape is a powerful tool for investigating the complex links between morphology and evolution, and their forcing factors over geological timescales. These new results complement those from the analysis of molecular data derived from extant taxa. Combining both results is likely to yield a clear picture on the evolutionary history of mammals and, specifically, on the forcing factors shaping their diversity and evolution.

## Methods

### Sample composition, data acquisition and digitization
Our research complies with all relevant ethical regulations. We selected and scanned petrosal bones, which house the bony labyrinth, from 306 specimens representing 191 ruminant species from the early Oligocene (ca. 33 Ma) to the present (Supplementary Data 2). The resulting dataset encompasses the most extensive study to date using

original data of the inner ear region representing 16% of the all known ruminant diversity including 16.5% of the known Tragulidae and ca. 22% of the known Bovidae and Cervidae. The following ruminant clades are included in the analysis, and for each pecoran lineage one of the earliest representatives is included (Supplementary Data 2): Antilocapridae, Bovidae, Cervidae, Dromomerycidae, Giraffomorpha *sensu*[55], Moschidae, as well as Middle Miocene Tragulidae and several members of Stem Pecora and Stem Ruminantia (Fig. 4 and Supplementary Fig. 3). More specific information concerning the specimens (taxonomical group, inventory number, host institution, age, and locality) is given in Supplementary Data 2.

Petrosal bones were scanned using high resolution hard X-ray computed tomography from the following institutions: Biomaterials Science Center of the University of Basel (CH), nanotom® m (phoenix| x-ray, GE Sensing & Inspection Technologies); Department of Anthropology of the University of Zurich (CH), Nikon XTH 225 ST; Department of Geosciences of the University of Fribourg (CH), Bruker Skyscan 2211; Plateforme d'Accès Scientifique à la Tomographie à Rayon X (AST-RX) of the Muséum national d'Histoire naturelle in Paris (FR), GE Sensing and Inspection Technologies phoenix X-ray v|tome|x L240-180; Plateforme Montpellier Ressources Imagerie (MRI) of the University of Montpellier 2 (FR), Skyscan 1076 in vivo; Microscopy and Imaging Facility (MIF) of the American Museum of Natural History (USA), GE Phoenix Vtome x L240; Nanoscale Research Facility of the University of Florida (USA), Phoenix v|tome|x M (GE's Measurement & Control business); Staatliche Naturwissenschaftliche Sammlungen Bayerns (G), nanotom® m (phoenix|x-ray, GE Sensing & Inspection Technologies); Staatliches Museum für Naturkunde Stuttgart (G), Bruker Skyscan 1272; The Natural History Museum London of United Kingdom (UK), Nikon Metrology HMX-ST 225; Institute of Vertebrate Paleontology and Paleoanthropology, Chinese Academy of Sciences (CHINA), GE v|tome|x m300&180 (GE Measurement & Control, Wuntsdorf, Germany); Museo Nacional de Ciencias Naturales-CSIC (ES), NIKON CT-SCAN- XT H-160; University Museum, University of Tokyo (JA), TX225-ACTIS (TESCO Corporation) and ScanXmate-B100TSS110 (Comscantecno Co. Ltd.). Pixel resolution mostly varies between 15 and 60 μm. During petrosal bone acquisition, 1440 equiangular radiographs were taken over 360° using an adjustable range of accelerating voltage of 90 kV and a beam current of 200 μA for recent material to 180 kV with a beam current of 30 μA for fossils. Segmentation was performed with AVIZO® 9.0 Lite software (FEI Visualization Sciences Group, Houston).

Digitization of the bony labyrinth was performed using Landmark Editor 3.6 software[56]. The landmarks data are available in Supplementary Data 1. The landmarking protocol has been improved from Mennecart et al[10]. (see Supplementary Fig. 5). The primary semilandmark curves were resampled in *R* v4.1.3[57] to produce an equidistant repartition of 307 points along the curves[58] using the "*digit.curves*" function of the *R* package *geomorph* v4.0.3[59,60]. The semi-landmarks were then slid along the curves using the bending energy[61].

### Phylogeny
A phylogenetic tree was obtained using Mesquite 3.04 software[62] combining specific phylogenetic hypotheses of the 191 species into a combined tree (*Parabos*[63], *Leptobos*[64], Antilocapridae[52], Dromomerycidae[65], Reduncini[66], *Myotragus*[67], duikers[68], extant Ruminantia[28], Stem Ruminantia and Stem Pecora[69,70], Giraffomorpha[55], Moschidae[71], Cervidae[72], Giraffidae[73]). We retain the familial topology where Antilocapridae is the sister clade of all other extant pecorans, Giraffomorpha is the sister clade of Cervidae and Bovoidea, and Bovoidea is composed of Moschidae and Bovidae[28]. Giraffomorpha contains Giraffidae, Palaeomerycidae, and Climacoceratidae[55]. The giraffid from Kohfidisch (Turolian; MN11) is currently described as Giraffidae gen. indet[74]. However, the morphology of the bony labyrinth shows typical structures that recall those observed in the *Birgerbohlinia schaubi* specimens of similar

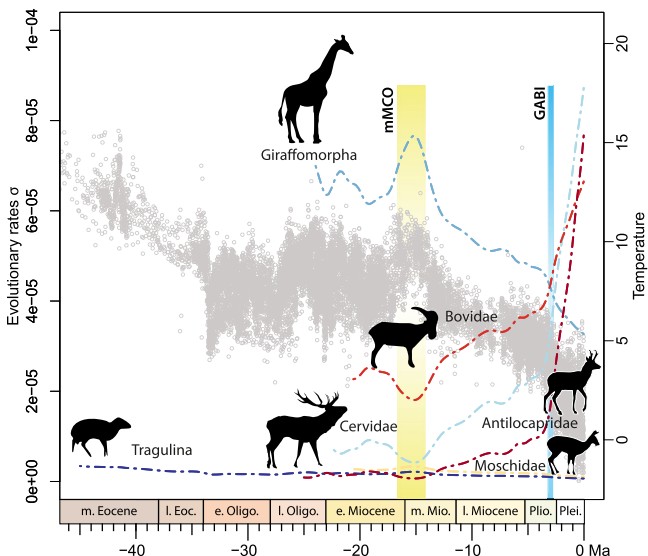

**Fig. 3 | Evolutionary rates of the different ruminant families' bony labyrinth morphology based on PC scores compared with the global temperature curve[1] (gray dots).** The Middle Miocene Climatic Optimum (mMCO) and the Great American Biotic Interchange (GABI) are highlighted since they are important factors in the ruminant evolution. Methodology and statistical results produced by the *R* package *RPanda*[93] are provided in Material and Methods and Supplementary Data 1. Groups are Tragulina (dark blue; Stem Ruminantia + Tragulidae), Antilocapridae (dark red), Giraffomorpha (dark light blue), Cervidae (light blue), Moschidae (yellow), Bovidae (red). Silhouettes of the families modified from[107].

age from the Crevillente-2 locality (Turolian; MN11; 8.5 Ma[75]). Nevertheless, based on several morphological differences (the thickness of the first cochlear turn, orientation of the vestibular aqueduct, and the anterior canal shape), we cannot confirm that Giraffidae gen. indet. from Kohfidisch is *Birgerbohlinia schaubi*. We attribute this specimen to *Birgerbohlinia* sp., sister taxon of *Birgerbohlinia schaubi*. Dromomerycidae has been considered to be closely related to Cervidae[55]. However, we retain the phylogenetic hypothesis of a close relation to Antilocapridae[76] based on morphological similarities of the bony labyrinth. *Hoplitomeryx* is a vigorously discussed taxon from a phylogenetic viewpoint for which a new family has been proposed[77]. Studies of the horncores of *Hoplitomeryx* show bovid affinity for this taxon[76,78]. We keep this phylogenetic hypothesis here, because the morphology of the bony labyrinth, especially the lateral semicircular canal orientation, is typical for bovids.

The nodes of the main clades are calibrated using the fossil record. The oldest known ruminant is *Archaeomeryx* from the middle Eocene of Asia, ca. 44 Ma[79]. The origin of the crown Pecora is older than 37 Ma[39]. The giraffomorph *Bedenomeryx* is known from 24 Ma[80]. The oldest moschid is ca. 18 Ma[81], while the oldest bovid ca. 18.75 Ma. Node calibration within families is estimated based on several sources in the literature[10,37,55]. The phylogenetic tree can be consulted in Supplementary Fig. 6 and the nexus file is given in the Supplementary Data 1.

### Statistical analysis
**Shape variation and phylogenetic signal.** Shape variation in bony labyrinth morphology (disparity and similarity) was studied using a geometric morphometrics approach implemented in MorphoJ[82] and *R* v4.1.3[57]. When several specimens of a species were available, we created a mean shape for the species using the function "*mshape*" of the package *geomorph* v.4.0.3[59,60] for the analyses that are using only one specimen per species in a phylogenetic framework (dataset with 191 species). We performed a Principal Component Analysis (PCA) computed using the function "*procSym*" of the *R* package *Morpho* v2.9[83] to study the shape variation within the dataset in its natural scale, including intraspecific variation (dataset with 306 specimens). In

order to assess our expectation that there should be a strong phylogenetic signal in our data, we performed a permutation test (randomized rounds: 10.000[84]) based on the phylogenetic tree. Klingenberg and Gidaszewski[84] defined that "The empirical *p-value* for the test is the proportion of permuted data sets in which the sum of squared changes is shorter or equal to the value obtained for the original data." Marriott[85] and Edgington[86] suggested that 1.000 permutations are a reasonable minimum for a test at 5% level of significance, while 5.000 are a reasonable minimum at the 1% level[87]. Moreover, the Pagel's λ and Bloomberg's K phylogenetic signal values have been calculated using the function "*phylosig*" in the *R* package *phytools* v1.0.3[88]. All supporting data are given in the Supplementary Data 1. To characterize the morphological similarities within the different clades, a between-groups PCA (bg-PCA) and a Canonical Variates Analysis (CVA) were performed. bg-PCA and CVA provide complementary information[89,90]. A bg-PCA observes the variance between groups (here defined as Stem Ruminantia, Tragulidae, Stem Pecora, Antilocapridae, Giraffomorpha, Cervidae, Moschidae, and Bovidae) without standardizing the within-groups variance[89]. A standardization is performed when using the CVA. The CVA maximizes the separation of the between-groups means relative to the variation within the groups' ratio according to the specified chosen grouping variable[89]. The bg-PCA was computed using the function "*groupPCA*" and the CVA using the function "*CVA*", both in the *R* package *Morpho* v2.9[83]. To test the performance of the classification model, both analyses were cross-validated using leaving-one-out cross-validation for the bg-PCA and Jackknife Cross-validation for the CVA. All supporting data of the bg-PCA and the CVA are given in Supplementary Data 1 (script) and in Supplementary Data 1 (supporting information).

**Evolutionary rate.** The evolutionary rate of the bony labyrinth morphology within the phylogeny was calculated using the function "*RRphylo*" in the *R* package *RRphylo* v2.6.0[91]. This function performs a phylogenetic ridge regression[92]. The significance of the evolutionary rate changes (acceleration and deceleration) were tested using function "*search.shift*" in the *R* package *RRphylo* v2.6.0[91] to highlight the significant changes in the rates within the phylogenetic tree. We plot the calculated evolutionary rate values on the phylogenetic tree using the function "*contMap*" in the *R* package *phytools* v1.0.3[88]. All supporting data concerning *RRphylo* are given in Supplementary Data 1.

We tested whether the bony labyrinth morphology evolutionary rate is correlated, anticorrelated, or not correlated (β) with an environmental function through time using the function "*fit_t_env*" in the *R* package *RPANDA* v2.0[93]. The environmental function is the isotopic data of the δ[18]O used as a proxy for the temperature curve provided by Zachos et al[1]. Because each family has a potentially different ecological optimum and reacts differently to the environmental changes, we separately analyzed each pecoran clade and the Tragulina in a first analysis. We then analyzed the Pecora as a whole and the Tragulina in a second analysis. The significance of the results was tested against different levels of smoothing of the temperature curve, compared to the resulting Aikake values. This implies that if these correlations are always significant in a clade, regardless of the smoothing parameter, temperature is not the only parameter influencing evolutionary rates. All supporting data concerning *RPANDA* are given in Supplementary Data 1.

To compare the impact of the topology of the phylogenetic tree on the evolutionary results, similar analyses have also been performed using the size of the specimens based on the bony labyrinth centroid size. Bony labyrinth size has been proven to be a good proxy for body mass estimation in ruminants[94] and has been used here to explore size evolution and diversification in ruminants. Bony labyrinth morphology is constrained by

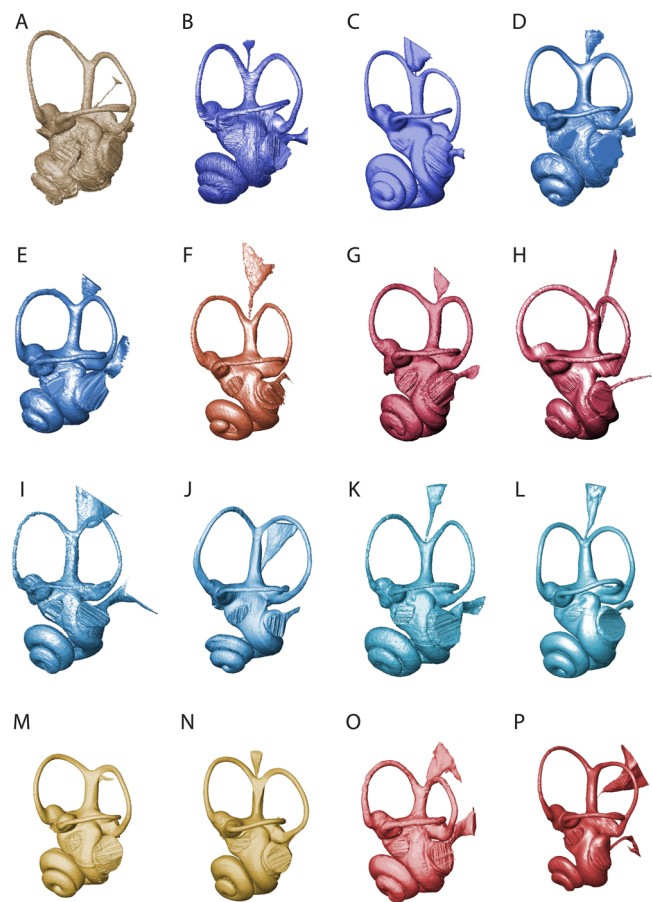

**Fig. 4 | Bony labyrinth morphology of the different groups of ruminants.**
This includes the bony labyrinth of the oldest currently known genera of all the
extant pecoran families, shown in ventrolateral view: Stem Ruminantia
(**A** *Hypisodus minimus* AMNH9354), Tragulidae (**B** *Dorcatherium crassum* NMB
San15053 and **C** *Moschiola meminna* NMB 2319), Stem Pecora (**D** *Prodremotherium
elongatum* MNHN.Qu4596 and **E** *Parablastomeryx primus* AMNH13822), Dromo-
merycidae (**F** *Dromomeryx scotti* AMNH FAM33800), Antilocapridae (**G** *Cosoryx
furcatus* AMNH FAM32426 and **H** *Antilocapra americana* NMB C.1618), Gir-
affomorpha (**I** *Ampelomeryx ginsburgi* Beon91G4 261 and **J** *Okapia johnstoni* NMB
10811), Cervidae (**K** *Procervulus dichotomus* SNSB-BSPG1979XV555 and **L** *Cervus
elaphus* NMB11147), Moschidae (**M** *Micromeryx flourensianus* NMB Sth.825 and
**N** *Moschus moschiferus* NMB 4201), and Bovidae (**O** *Eotragus artenensis* SMNS50Mu
and **P** *Capra ibex* NMB 5837).

Evolutionary rates of the bony labyrinth are more heterogenous
and explained in detail in the text.

## Reporting summary

Further information on research design is available in the Nature
Portfolio Reporting Summary linked to this article.

## Data availability

The datasets generated during and analyzed during the current study
(Landmark data) are available in the Supplementary Data 1. All mate-
rials from which shape data are generated are housed in museum
collections. Details on the location of these collections can be found in
Supplementary Data 2. 3D reconstructed models of the bony labyr-
inths are either published and open acces in MorphoMuseuM[102–106]
(https://morphomuseum.com/) or will be. The not yet published
models are available from the corresponding authors upon reasonable
request.

## Code availability

The code generated to conduct this study is available in the GitHub
repository https://github.com/SilberdistL/inner-ear_Ruminants.git and
is archived on Zenodo (https://doi.org/10.5281/zenodo.7060117).

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

labyrinth morphology alone. The results of these analyses are
given in Supplementary Figs. 7, 8, and Supplementary Data 1. We
find that evolution of size differs distinctively from bony labyr-
inth morphological evolution (evolutionary rate and significant
shifts in Supplementary Figure 7, as well as large-scale correlation
with environmental factors; Supplementary Data 1). This indicates
that the topology of the phylogenetic tree and the number of
considered taxa are not the main factors driving our results.
Increases in maximum body size and the evolutionary rate at
which size increase can be observed through time in most of the
clades (Supplementary Fig. 8). The tendency of evolutionary
lineages to increase their body size through time is known as the
Depéret-Cope's rule[100]. An increase in the evolutionary rates of
size is known from insular contexts and during radiations[101].

15. Costeur, L. et al. The bony labyrinth of toothed whales reflects both phylogeny and habitat preferences. *Sci. Rep.* **8**, 7841 (2018).

16. Spoor, F., Bajpai, S., Hussain, S. T., Kumar, K. & Thewissen, J. G. M. Vestibular evidence for the evolution of aquatic behavior in early cetaceans. *Nature* **417**, 163–166 (2002).

17. Davies, K. T. J., Bates, P. J. J., Maryanto, I., Cotton, J. A. & Rossiter, S. J. The evolution of bat vestibular systems in the face of potential antagonistic selection pressures for flight and echolocation. *PLoS ONE* **8**, e61998 (2013).

18. Park, T., Mennecart, B., Costeur, L., Grohé, C. & Cooper, N. Convergent evolution in toothed whale cochleae. *BMC Evol. Biol.* **1**, 195 (2019).

19. Benoit, J. et al. A test of the lateral semicircular canal correlation to head posture, diet and other biological traits in "ungulate" mammals. *Sci. Rep.* **10**, 19602 (2020).

20. Morimoto, N. et al. Variation of bony labyrinthine morphology in Mio-Plio-Pleistocene and modern anthropoids. *Am. J. Phys. Anthropol.* **2020**, 1–17 (2020).

21. DeMiguel, D., Azanza, B. & Morales, J. Key innovations in ruminant evolution: A paleontological perspective. *Int. Zool.* **9**, 412–433 (2014).

22. Gunz, P., Ramsier, M., Kuhrig, M., Hublin, J. & Spoor, F. The mammalian bony labyrinth reconsidered, introducing a comprehensive geometric morphometric approach. *J. Anat.* **220**, 529–543 (2012).

23. Grohe, C., Tseng, Z. J., Lebrun, R., Boistel, R. & Flynn, J. J. Bony labyrinth shape variation in extant Carnivora: a case study of Musteloidea. *J. Anat.* **228**, 366–383 (2015).

24. Urciuoli, A. et al. A comparative analysis of the vestibular apparatus in *Epipliopithecus vindobonensis*: Phylogenetic implications. *J. Hum. Evol.* **151**, 102930 (2021).

25. IUCN. The IUCN red list of threatened species. Version 2021-1. https://www.iucnredlist.org. Accessed 17 June 2021.

26. Kingdon, J. & Hoffmann. M. *Mammals of Africa. Volume VI pigs, hippopotamuses, chevrotains, Giraffes, deer and bovids* 704 (Bloomsbury Publishing, 2013).

27. Chen, L. et al. Large-scale ruminant genome sequencing provides insights into their evolution and distinct traits. *Science* **364** eaav6202 (2019).

28. Hassanin, A. et al. Pattern and timing of diversification of Cetartiodactyla (Mammalia, Laurasiatheria), as revealed by a comprehensive analysis of mitochondrial genomes. *C. R. Biol.* **335**, 32–50 (2012).

29. Wang, Y. et al. Genetic basis of ruminant headgear and rapid antler regeneration. *Science* **364**, 1153 (2019).

30. Myers, E. A. & Bubrink, F. T. Ecological opportunity: Trigger of adaptative radiation. *Nat. Educ. Knowl.* **3**, 23 (2012).

31. Gentry, A. W. Bovidae. In *Cenozoic mammals of Africa* (eds Werdelin, L. & Sanders, W. J.) 741–796 (University of California Press, 2010).

32. Harris, J. M., Solounias, N. & Geraads, D. Giraffoidea. In Werdelin, L. & Sanders, W. J. *Cenozoic mammals of Africa*. 797–812 (University of California Press, 2010).

33. Clauss, M. & Rössner, G. E. Old world ruminant morphophysiology, life history, and fossil record: exploring key innovations of a diversification sequence. *Ann. Zool. Fenn.* **51**, 80–94 (2014).

34. Johnston, A. R. & Anthony, N. M. A multi-locus species phylogeny of African forest duikers in the subfamily Cephalophinae: evidence for a recent radiation in the Pleistocene. *BMC Evol. Biol.* **12**, 120 (2012).

35. Cooney, C. R. & Thomas, G. H. Heterogeneous relationships between rates of speciation and body size evolution across vertebrate clades. *Nat. Ecol. Evol.* **5**, 101–110 (2020).

36. Köhler, M. & Moyà-Solà, S. Physiological and life history strategies of a fossil large mammal in a resource-limited environment. *Proc. Natl Acad. Sci. USA* **106**, 20354–22035 (2009).

37. Bibi, F. A multi-calibrated mitochondrial phylogeny of extant Bovidae (Artiodactyla, Ruminantia) and the importance of the fossil record to systematics. *BMC Evol. Biol.* **13**, 1–15 (2013).

38. Geraads, D. A reassessment of the Bovidae (Mammalia) from the Nawata Formation of Lothagam, Kenya, and the late Miocene diversification of the family in Africa. *J. Syst. Palaeontol.* **17**, 1–14 (2017).

39. Mennecart, B., Aiglstorfer, M., Li, Y., Li, C. & Wang, S. Ruminants reveal Eocene Asiatic palaeobiogeographical provinces as the origin of diachronous mammalian Oligocene dispersals into Europe. *Sci. Rep.* **11**, 17710 (2021).

40. Rössner, G. E. Family tragulidae. In: *The evolution of artiodactyls* (eds Prothero, D. R. & Foss S. C.) (The Johns Hopkins University Press, Baltimore, 2007).

41. Sánchez, I. M., Quiralte, V., Morales, J. & Pickford, M. A new genus of tragulid ruminant from the early Miocene of Kenya. *Acta Palaeontol. Pol.* **55**, 177–187 (2010).

42. Sánchez, I. M., Quiralte, V., Ríos, M., Morales, J. & Pickford, M. First African record of the Miocene Asian mouse-deer *Siamotragulus* (Mammalia, Ruminantia, Tragulidae): implications for the phylogeny and evolutionary history of the advanced selenodont tragulids. *J. Syst. Palaeontol.* **13**, 543–556 (2015).

43. Mennecart, B. et al. The first French tragulid skull (Mammalia, Ruminantia, Tragulidae) and associated tragulid remains from the Middle Miocene of Contres (Loir-et-Cher, France). *C. R. Palevol* **17**, 189–200 (2018).

44. Bobe, R. & Eck, G. C. Responses of African bovids to Pliocene climatic change. *Paleobiology* **27**, 1–47 (2001).

45. Strömberg, C. A. E. Decoupled taxonomic radiation and ecological expansion of open-habitat grasses in the Cenozoic of North America. *Proc. Natl Acad. Sci. USA* **102**, 11980–11984 (2005).

46. Kaya, F. et al. The rise and fall of the Old World savannah fauna and the origins of the African savannah biome. *Nat. Ecol. Evol.* **2**, 241–246 (2017).

47. Gravilets, S. & Losos, J. B. Adaptive radiation: contrasting theory with data. *Science* **323**, 732–737 (2009).

48. Moen, D. & Morlon, H. Why does diversification slow down? *Trends Ecol. Evol.* **29**, 190–197 (2014).

49. Couvreur, T. L. P. et al. Tectonics, climate and the diversification of the tropical African terrestrial flora and fauna. *Biol. Rev.* **96**, 16–51 (2020).

50. Fontoura, E., Darival Ferreira, J., Bubadué, J., Ribeiro, A. M. & Kerber, L. Virtual brain endocast of *Antifer* (Mammalia: Cervidae), an extinct large cervid from South America. *J. Morphol.* **281**, 1–18 (2020).

51. Trauth M. A. et al. Recurring types of variability and transitions in the ~620 kyr record of climate change from the Chew Bahir basin, southern Ethiopia Quaternary. *Sci. Rev.* https://doi.org/10.1016/j.quascirev.2020.106777 (2021).

52. Janis, C. M. & Manning, E. Antilocapridae. In *Evolution of tertiary mammals of North America* (eds Janis, C. M., Scott, K. M. & Jacobs, L. L.) 491–507 (Cambridge University Press, 1998).

53. Klimova, A., Munguia-Vega, A., Hoffman, J. I. & Culver, M. Genetic diversity and demography of two endangered captive pronghorn subspecies from the Sonoran Desert. *J. Mammal.* **95**, 1263–1277 (2014).

54. Evin, A., et al. Size and shape of the semicircular canal of the inner ear: A new marker of pig domestication? *J. Exp. Zool. B Mol. Dev. Evol.* https://doi.org/10.1002/jez.b.23127 (2022).

55. Sánchez, I. M., Cantalapiedra, J. L., Ríos, M., Quiralte, V. & Morales, J. Systematics and evolution of the Miocene three-horned Palaeomerycid ruminants (Mammalia, Cetartiodactyla). *PLoS ONE* **10**, e0143034 (2015).

56. Wiley, D. *Landmark Editor 3.6* (Institute for Data Analysis and Visualization, Davis, CA, University of California, 2006).

57. R Core Team. *R: A language and environment for statistical computing* (R Foundation for Statistical Computing, Vienna, Austria, 2022). https://www.R-project.org/.

58. Gunz, P. & Mitteroecker, P. Semilandmarks: a method for quantifying curves and surfaces. *Hystrix* **24**, 103–109 (2013).

59. Adams, D. C. & Otárola-Castillo, E. geomorph: an R package for the collection and analysis of geometric morphometric shape data. *Methods Ecol. Evol.* **4**, 393–399 (2013).

60. Adams, D. C., Collyer, M. L., Kaliontzopoulou, A. *geomorph: software for geometric morphometric analyses*. R package version 3.2.1 software (2020).

61. Gunz, P., Mitteroecker, P., Bookstein, F. L. Semilandmarks in three dimensions. In Modern morphometrics in physical anthropology. Springer, pp. 73–98 (2005).

62. Maddison, W. P., Maddison, D. R. *Mesquite: a modular system for evolutionary analysis*. Version 3.04. (2010).

63. Gromolard, C. & Guérin, C. Mise au point sur *Parabos cordieri* (de Christol), un Bovidé (Mammalia, Artiodactyla) du Pliocène d'Europe occidentale. *Géobios* **13**, 741–755 (1980).

64. Duvernois, M.-P. Mise au point sur le genre *Leptobos* (Mammalia, Artiodactyla, Bovidae); implications biostratigraphiques et phylogénétiques. *Géobios* **25**, 155–166 (1992).

65. Janis, C. M., Manning, E. Dromomerycidae. In *Evolution of Tertiary mammals of North America Volume1: Terrestrial carnivores, ungulates, and ungulatelike mammals* (eds. Janis, C. M., Scott, K. M., Jacobs L. L.) 477–490 (Cambridge University Press, 1998).

66. Birungi, J. & Arctander, P. Molecular systematics and phylogeny of the reduncini (artiodactyla: bovidae) inferred from the analysis of mitochondrial cytochrome b gene sequences. *J. Mamm. Evol.* **8**, 125–147 (2001).

67. Lalueza-Fox, C. et al. Molecular dating of caprines using ancient DNA sequences of *Myotragus balearicus*, an extinct endemic Balear mammal. *BMC Evol. Biol.* **5**, 1–11 (2005).

68. Marot, J. D. Molecular phylogeny of terrestrial artiodactyls, conflict and resolution. In *The evolution of artiodactyls* (eds Prothero, D. R., Foss, S. C.) 4–18 (The Johns Hopkins University Press, 2007).

69. Webb, D. S. Hornless ruminants. In *Evolution of Tertiary mammals of North America Volume1: Terrestrial carnivores, ungulates, and ungulatelike mammals* (eds Janis, C. M., Scott, K. M., Jacobs, L. L.) 463–476 (Cambridge University Press, 1998).

70. Mennecart, B. & Métais, G. *Mosaicomeryx* gen. nov., a ruminant mammal from the Oligocene of Europe and the significance of 'gelocids'. *J. Syst. Palaeontol.* **13**, 581–600 (2015).

71. Sánchez, I. M., DeMiguel, D., Quiralte, V. & Morales, J. The first known Asian Hispanomeryx (Mammalia, Ruminantia, Moschidae.). *J. Vert. Paleontolo.* **31**, 1397–1403 (2011).

72. Heckeberg, N. S., Erpenbeck, D., Wörheide, G. & Rössner, G. Systematic relationships of five newly sequenced cervid species. *PeerJ* **4**, e2307 (2016).

73. Ríos, M., Sánchez, I. M. & Morales, J. A new giraffid (Mammalia, Ruminantia, Pecora) from the late Miocene of Spain, and the evolution of the sivathere-samothere lineage. *PLoS ONE* **12**, e0185378 (2017).

74. Vislobokova, I. New data on late Miocene mammals of Kohfidisch, Austria. *Paleontol. J.* **41**, 451–460 (2007).

75. Aiglstorfer, M., Rössner, G. E. & Böhme, M. *Dorcatherium naui* and pecoran ruminants from the late Middle Miocene Gratkorn locality (Austria). *Palaeobiodivers. Palaeoenviron.* **94**, 83–123 (2014).

76. Janis, C. M. & Scott, K. M. The interrelationships of higher ruminant families with special emphasis on the members of the Cervoidea. *Am. Mus. Novit.* **2893**, 1–85 (1987).

77. Leinders, J. Hoplitomerycidae fam. nov. (Ruminantia, Mammalia) from Neogene fissure fillings in Gargano (Italy). *Scr. Geol.* **70**, 1–68 (1984).

78. Hassanin, A. & Douzery, E. Molecular and morphological phylogenies of Ruminantia, and the alternative position of the Moschidae. *Syst. Biol.* **52**, 206–228 (2003).

79. Métais, G. & Vislobokova, I. Basal ruminants. In *The evolution of artiodactyls* (eds Prothero, D. R. & Foss, S. C.) 189–212 (The Johns Hopkins University Press, 2007).

80. Mennecart, B., Zoboli, D., Costeur, L. & Pillola, G. L. On the systematic position of the oldest insular ruminant *Sardomeryx oschiriensis* (Mammalia, Ruminantia) and the early evolution of the Giraffomorpha. *J. Syst. Palaeontol.* **17**, 691–704 (2019).

81. Aiglstorfer, M. et al. Musk Deer on the Run - Dispersal of Miocene Moschidae in the Context of Environmental Changes. In *Evolution of Cenozoic land mammal faunas and ecosystems: 25 years of the NOW database of fossil mammals*. (eds Casanovas-Vilar, I., van den Hoek Ostende, L. W., Janis, C. M. & Saarinen J.) (Cham: Springer, in press).

82. Klingenberg, C. P. MorphoJ: an integrated software package for geometric morphometrics. *Mol. Ecol. Resour.* **11**, 353–357 (2011).

83. Schlager, S. Morpho and Rvcg – Shape analysis in R. In Zheng, G., Li, S., Szekely, G. *Statistical shape and deformation analysis*, 217–256 (MA: Academic Press, 2017).

84. Klingenberg, C. P. & Gidaszewski, N. A. Testing and quantifying phylogenetic signals and homoplasy in morphometric data. *Syst. Biol.* **59**, 245–261 (2010).

85. Marriott, F. H. C. Barnard's monte carlo tests: how many simulations? *Appl. Stat.* **28**, 75–77 (1979).

86. Edgington, E. S. *Randomization tests* (Marcel Dekker, 1987).

87. Tzeng, T. D. & Yeh, S. Y. Permutation tests for difference between two multivariate allometric patterns. *Zool. Stud.* **38**, 10–18 (1999).

88. Revell, L. J. phytools: an R package for phylogenetic comparative biology (and other things). *Methods Ecol. Evol.* **3**, 217–223 (2012).

89. Renaud, S., Dufour, A.-B., Hardouin, E. A., Ledevin, R. & Auffray, C. Once upon multivariate analyses: when they tell several stories about biological evolution. *PLoS ONE* **10**, e0132801 (2015).

90. Mitteroecker, P. & Bookstein, F. Linear discrimination, ordination, and the visualization of selection gradients in modern morphometrics. *Evol. Biol.* **38**, 100–114 (2011).

91. Raia, P., Castiglione, S., Serio, C., Mondanaro, A. & Raia, M. P. Package 'RRphylo'. *CRAN Repos.* **4**, 1–31 (2018).

92. Castiglione, S. et al. A new method for testing evolutionary rate variation and shifts in phenotypic evolution. *Methods Ecol. Evol.* **9**, 974–983 (2018).

93. Morlon, H. et al. "RPANDA: an R package for macroevolutionary analyses on phylogenetic trees.". *Methods Ecol. Evol.* **7**, 589–597 (2016).

94. Costeur, L., Mennecart, B., Müller, B., Schulz, G. Observations on the scaling relationship between bony labyrinth, skull size and body mass in ruminants. *Proc. SPIE* **11113**, https://doi.org/10.1117/12.2530702 (2019).

95. Costeur, L., Mennecart, B., Müller, B. & Schulz, G. Prenatal growth stages show the development of the ruminant bony labyrinth and petrosal bone. *J. Anat.* **230**, 347–353 (2017).

96. Mennecart, B. & Costeur, L. Shape variation and ontogeny of the ruminant bony labyrinth, an example in Tragulidae. *J. Anat.* **229**, 422–435 (2016).

97. Clauss, M., Steuer, P., Müller, D. W. H., Codron, D. & Hummel, J. Herbivory and body size: allometries of diet quality and gastro-intestinal physiology, and implications for herbivore ecology and dinosaur gigantism. *PLoS One* **8**, e68714 (2013).

98. du Toit, J. T. & Owen-Smith, N. Body size, population metabolism, and habitat specialization among large African herbivores. *Am. Nat.* **133**, 736–740 (1989).

99. Mennecart B., Becker D., & Berger J. -P. Mandible shape of ruminants: between phylogeny and feeding habits. In: *Ruminants: Anatomy, behavior, and diseases*, (ed. Mendes R. E.) 205–226 (Nova Science Publishers, 2012).

100. Bokma, F. et al. Testing for Depéret's rule (body size increase) in mammals using combined extinct and extant data. *Syst. Biol.* **65**, 98–108 (2016).

101. Besiou, E., Choupa, M. N., Lyras, G. & van der Geer, A. Body mass divergence in sympatric deer species of Pleistocene Crete (Greece). *Palaeontol. Electron.* **25**, a23 (2022).

102. Mennecart B., Métais G., Tissier J., Rössner G. E., & Costeur L. 3D models related to the publication: Reassessment of the enigmatic ruminant Miocene genus *Amphimoschus* Bourgeois, 1873 (Mammalia, Artiodactyla, Ruminantia, Pecora). *MorphoMuseuM* **7**, e131 (2021).

103. Mennecart, B., Perthuis de, A. D. & Costeur, L. 3D models related to the publication: The first French tragulid skull (Mammalia, Ruminantia, Tragulidae) and associated tragulid remains from the Middle Miocene of Contres (Loir-et-Cher, France). *MorphoMuseuM* **3**, e4 (2018).

104. Aiglstorfer, M., Costeur, L., Mennecart, B. & Heizmann, E. P. J. *Micromeryx? eiselei* - a new moschid species from Steinheim am Albuch, Germany, and the first comprehensive description of moschid cranial material from the Miocene of Central Europe. *MorphoMuseuM* **3**, e4 (2107).

105. Costeur, L. & Mennecart, B. 3D models related to the publication: Prenatal growth stages show the development of the ruminant bony labyrinth and petrosal bone. *MorphoMuseuM* **2**, e3 (2016).

106. Mennecart, B. & Costeur, L. 3D models related to the publication: a *Dorcatherium* (Mammalia, Ruminantia, Middle Miocene) petrosal bone and the tragulid ear region. *MorphoMuseuM* **2**, e2 (2016).

107. Mennecart, B. et al. Allometric and phylogenetic aspects of stapes morphology in ruminantia (Mammalia, Artiodactyla). *Front. Earth Sci.* **8**, 176 (2020).

## Acknowledgements

We would like to thank all the curators, collection manager, and scientists who helped us allowed us to study and provided access to material: C. Argot and G. Billet (Muséum national d'Histoire naturelle, Paris), D. Geraads (Museum national d'Histoire Naturelle) for providing the *Sivatherium* specimen, E. Robert (Université Claude Bernard, Lyon 1), D. Berthet (Musée des Confluences Lyon), M. Orliac (Université Montpellier 2) for providing the bony labyrinth of *Bachitherium*, F. Duranthon and Y. Laurent (Muséum d'histoire naturelle, Toulouse), S. Legal, P. Coster, C. Balm, O. Maridet, O. Lapauze and J. Tissier (Parc Naturel Régional du Luberon and excavation team Murs Project), A. de Perthuie for accessing his private collection, Christiane Zeitler, R. Ziegler and E. Heizmann (Staatliches Museum für Naturkunde Stuttgart), P. Brewer, A. Garbout, and F. Ahmed (Natural History Museum, London), U. Göhlich and G. Daxner-Höck (Naturhistorisches Museum Wien), A. Van der Geer (Netherland Centre for Biodiversity Leiden), R. C. Hulbert Jr. (University of Florida, Gainesville), A. M. García Forner and P. Montoya (Museu de Geologia de la Universitat de València, Burjassot) for providing the petrosals of *Birgerbohlinia*, M. Pina (University of Manchester) for scanning support, and J. Morales (Museo Nacional de Ciencias Naturales, Madrid), J. Galkin, R. O'Leary, M. Hill Chase, C. Grohé and A. Gishlick (AMNH New York, USA), M. Celik (University of Queensland), and M. Scheidegger. We are also grateful to all the persons and institutions who scanned for us. Tandra Fairbanks (NMB) is thanked for her help with the English. B.M. and L.C. are grateful to the Swiss National Science Foundation for supporting this research through the projects 200021_178853 and 200021_159854/1 on the ear region evolution in ruminants. B.M. P300P2_161065 and P3P3P2_161066 on the evolution of the early ruminants. F.B. acknowledges support from a Gerstner Scholarship at the American Museum of Natural History. G.R. thanks the German Research Foundation project RO 1197/3-1. G.M., B.M., and L.C. want to thank the Museum National d'Histoire Naturelle de Paris for financing the Ast-RX-2013-051 Project. D.D.M. acknowledges R+D +I project ref. PID2020-116220GB-I00 from the Ministerio de Ciencia e Innovación/Agencia Estatal de Investigación/10.13039/ 501100011033/. I.S. acknowledges support by the Spanish Ministry of Science and Innovation (projects ref. PID2020-117289GB-I00 and PID2020-116220GB-I00), and the Generalitat de Catalunya (CERCA Programme). M.K. is funded by Japan Society for the Promotion of Science (KAKENHI Grant No. 19K04060). M.R. is supported by FCT-CEEC postdoctoral funding (CEECIND/02199/2018). M.R. acknowledges MINECO project CGL2011-25754 for providing funding for the analysis. S.W. thanks the Strategic Priority Research Program of Chinese Academy of Sciences (XDB26000000) and the National Natural Science Foundation of China (41872001). G.S. and B.Mü. acknowledge financial support from the Swiss National Science Foundation in the frame of the R'equip initiative (316030_133802).

## Author contributions

All the authors have approved the submitted version (and any substantially modified version that involves the author's contribution to the study); AND to have agreed both to be personally accountable for the author's own contributions and to ensure that questions related to the accuracy or integrity of any part of the work, even ones in which the author was not personally involved, are appropriately investigated, resolved, and the resolution documented in the literature. Conception or design of the work: B.M., I.D., L.C.; the acquisition: B.M., I.D., M.A., F.B., D.D.M., M.F., M.K., F.L., J.M., G.M., B.Mü., M.R., G.R., I.S., G.S., S.W., L.C.; the analysis: B.M., I.D.; the interpretation of data: B.M., I.D., L.C.; have drafted the work: B.M., I.D., L.C.; substantively revised it: B.M., I.D., M.A., F.B., D.D.M., M.F., M.K., F.L., J.M., G.M., B.Mü., M.R., G.R., I.S., G.S., S.W., L.C.

## Competing interests

The authors declare no competing interests.

## Additional information

[1]Naturhistorisches Museum Basel, Augustinergasse 2, 4001 Basel, Switzerland. [2]Institute of Plant Sciences, University of Bern, 3013 Bern, Switzerland. [3]Oeschger Centre for Climate Change Research, University of Bern, 3012 Bern, Switzerland. [4]Naturhistorisches Museum Mainz / Landessammlung für Naturkunde Rheinland-Pfalz, Reichklarastraße 10, 55116 Mainz, Germany. [5]Museum für Naturkunde, Leibniz Institute for Evolution and Biodiversity Science, Berlin 10115, Germany. [6]Fundación ARAID, Zaragoza, Spain. [7]Departamento de Ciencias de la Tierra, Área de Paleontología / Instituto Universitario de Investigación en Ciencias Ambientales de Aragón (IUCA). Universidad de Zaragoza, Pedro Cerbuna 12, 50009 Zaragoza, Spain. [8]Institut Català de Palaeontologia Miquel Crusafont (ICP), Edifici Z, c/de les columnes s/n, Universitat Autònoma de Barcelona, 08193Cerdanyola del Vallès, Barcelona, Spain. [9]National Museum of Nature and Science, Tsukuba, Japan. [10]Department of Natural Environmental Studies, Graduate School of Frontier Sciences, The University of Tokyo, Chiba, Japan. [11]Swiss National Data and Service Center for the Humanities, 4123 Allschwil, Switzerland. [12]American Museum of Natural History, 10024 New York; Earth and Environmental Sciences, Graduate Center, City University of New York, New York, NY 10016, USA. [13]CR2P - Centre de Recherche en Paléontologie - Paris, UMR 7207, CNRS, MNHN, Sorbonne Université. Muséum national d'Histoire naturelle, CP38, 8 rue Buffon, 75005 Paris, France. [14]Biomaterials Science Center, Department of Biomedical Engineering, University of Basel, Gewerbestrasse 14, 4123 Allschwil, Switzerland. [15]Department of Earth Sciences, GeoBioTec, Nova School of Science and Technology, Universidade NOVA de Lisboa, Campus de Caparica, 2829-516 Caparica, Portugal. [16]Staatliche Naturwissenschaftliche Sammlungen Bayerns - Bayerische Staatssammlung für Paläontologie und Geologie, Richard-Wagner-Strasse 10, 80333 Munich, Germany. [17]Department für Geo- und Umweltwissenschaften, Paläontologie & Geobiologie, Ludwig-Maximilians-Universität München, Richard-Wagner-Strasse 10, 80333 Munich, Germany. [18]Micro- and Nanotomography Core Facility, Department of Biomedical Engineering, University of Basel Gewerbestrasse 14, 4123 Allschwil, Switzerland. [19]Institute of Vertebrate Paleontology and Paleoanthropology, Chinese Academy of Sciences, 142 Xizhimenwai Street, Beijing 100044, China. ✉e-mail: mennecartbastien@gmail.com

