## [Peer Review File · Nature Communications]

Ruminant inner ear shape records 35 million years of neutral evolutionReviewers' Comments:

Reviewer #1:

Remarks to the Author:

The manuscript titled "Hidden in the deep: ruminant inner ear sensory organs record 35 million years of major historical events" relies on shape changes occurring in the ruminant inner ear bony labyrinth for tracing the evolutionary history of this successful group of artiodactyls. To do so, the authors are relying on a previously described 3DGM approach (also slightly updated), which is applied to a very large sample of extant and fossil ruminants. The shape data is then analyzed by means of recently developed methods, which have not been applied to BL morphology before. Based on their results, the authors observe that changes in BL evolutionary rates occur in correspondence of major paleoecological events and conclude that BL morphology is well suited as a proxy for studying adaptive radiations. As a whole, this contribution appears well conceived and very comprehensive (especially sample-wise), with adequate descriptions of the methods and a massive amount of additional data, with the latter that is very useful for further supporting the conclusions reached, as well as to ensure the repeatability of the analyses. Thanks to the multidisciplinary approach employed, the novelty of the methods and the far-reaching conclusions that could be drawn, the results here presented constitute a remarkable advance in multiple fields related with evolutionary studies, and will be of interest to a large audience of both specialized and unspecialized readers.

The numerous positive aspects detailed above notwithstanding, there are a few points that would like to bring to the attention of the authors and ask them to consider.

Overall, a very interesting contribution
Alessandro Urciuoli

General comments:

- The overall format of the paper is rather unusual in my opinion. I understand that this is mostly an editorial matter, but the authors do not seem to have followed the format suggested by the journal neither in the abstract (>150 words long) nor in the sections in which the paper should be subdivided. Accordingly, I would suggest the authors to follow a more linear and "standard" structure, which should possibly include Introduction, Results and Discussion sections.
- Regardless of whether the authors would re-structure their paper, I would like to suggest them to provide a broader background to their study in which they should expand the various topics that they are already providing in the current draft e.g., why climate change is an important evolutionary factor? What has been done already in bony labyrinth studies? What is the state of the art about ruminant phylogeny and evolution? Which are the radiation events the authors are willing to trace? I think that this might be most useful for the readers, especially unspecialized ones, to obtain insights about why the authors are relying on ruminants and BL morphology to test their evolutionary hypotheses.
- Similar to the above, the authors in the current version are almost always directly discussing the results without previously present them to the reader. While the huge amount of supplementary data allows a correct interpretation, this requires a considerable amount of effort for the reader, which has to constantly switch between the main text and the supplementary data. This particularly applies to the different morphospaces here presented (PCA, bgPCA and CVA), which lack a proper description of the distribution of the taxa and of the patterns of shape variation observed along the considered axes (PC1-2, bgPC1-2, CV1-2). Since the current online content of the manuscript amounts to ~2000 words (with 5000 being the maximum), the authors should be able to include additional descriptions of their results and treat them separately in the discussion.

Specific comments

L49 – “historical parameters” I am not sure about the meaning the authors are intending with “historical”. Please clarify

L52 – “of the inner ear of the iconic ruminants” Please remove “the”

L60 – “fostered the development of heterogeneous environments promoting diversification and evolutionary rates in these clades” Possibly the authors omitted “fast” before “evolutionary rates”? Also, could the authors specify which are the clades they are referring to? Both deer and bovids, ruminants as a whole, or other ruminant clades?

L61-63 – I understand that the authors are willing one of the reasons for which they reached the conclusions mentioned in the period above. However, the current wording does not provide an adequate connection between the two periods. Please consider rephrasing.

L74 – “of modern and fossil mammals” is the previous statement restricted to mammals only? If it is the case, could the authors explain why and provide reference?

L75 – “An emerging anatomical object in evolutionary research that relates to the relationships (i.e. phylogeny) of animals”. The wording of this sentence is somewhat unclear, even if overall correct. I would suggest to the authors to consider the wording “An emerging anatomical object in evolutionary research thought to accurately reflect the kinship (i.e., phylogeny) between different species”

L79 – “outstanding analyses”. This is unclear to me. Could the authors clarify what are “outstanding analyses” in their view? Were they referring to recently developed methods?

L80 – “few” The use of this adjective is not correct. Please consider rephrasing.

L80-81 – “Then this structure may neutrally evolve like in genetic drift” Might authors expand on this concept? I understand that they are willing to highlight that the absence of hard functional constraints allowed the BL morphology to evolve neutrally. However, the wording appears slightly unclear and the connection with the previous sentence is not very easy to be followed.

L81-86 – Similarly to what happens with the previous sentence, this paragraph changes subject in a quite abrupt manner and does not connect appropriately with the previous one.

L91 – CT or microCT? The authors are previously referring to the latter.

L94 – “general” Generalist? Unspecialized? Primitive? General is too vague in my opinion. Please clarify.

L102-103 – Either “Pronghorn BL morphologies are the closest to those of the stem pecorans, while those of bovids are more derived,” or “Pronghorn BL morphology is the closest to that of the stem pecorans, while that of bovids is more derived”. Also, could the author provide information about why the observed morphologies are primitive or derived?

L161 – “limited” did the authors mean “limited to”?

L167 – “cooling reducing” possibly replace with “, which reduced”

L176-178 – I would suggest the authors to connect the two sentences with “where at least 19 extant species evolved locally and currently represent at least”

L178-182 – Very long sentence, consider splitting at least with commas or into separate sentences.

L184 – “stayed” and “little”, both words are not fitting entirely well in my opinion to the sentence and I would like to suggest the authors to consider replacing them with “remained” and “poorly”, respectively.

L184-185 – “and mostly inhabited rather open and similar habitats throughout their entire history” could this possibly be the cause of the low diversification? If so, the authors should be more explicit about it.

L185-188 – The authors here provide possible explanations about the very distinct morphology of Antilocapra relative to the other representative of the clade. However, and as specified above for the derived/primitive condition of bovids and pronghorns, in which terms is Antilocapra derived? The reader can indeed spot some differences between Fig. S3g and h (the authors could also directly refer to these frames in Fig. S3 for clarity) yet no general description of the bony labyrinth is provided so it seems somewhat unclear to me. In my opinion, the authors should provide a brief resume about which are the primitive features found in fossil antilocaprids, as well as the derived ones of extant

antilopatra. In addition to the above, does this statement hold true for more recent species such as Stockoceros and Capromeryx? I understand that the space constraints for the already very large Fig. S3 does not allow to add them, but the absence of the latter two taxa in the figure does not allow to ascertain this.

L200-208 – The authors are here delineating their conclusions, but the narration appears jumpy as the authors are changing topic in a somewhat abrupt way. I agree with them about the similar pattern observed in the evolution of body mass and BL shape. However, while reading this paragraph it is not clear, at first glance, why the authors are bringing up this here. The author could try to give a brief explanation (even better if this is done in the introduction) about why the BM evolutionary rates are a good proxy for inferring climatic fluctuations.

L203-205 – In my opinion the authors should consider rephrasing this period to clarify what is the evidence they found in their data and how this reflects the changes occurring in paleoecological factors. At present, it appears to be the other way around.

L203-208 – The periods found here in sequence do not connect adequately one to the other and are overall somewhat sketchy. Since these represent the take-home-message of the paper, I would like to suggest the authors to enhance their clarity.

L359 – The authors refer here to the range of voxel size for the microCT data. Could the authors provide pixel resolution and facility for each of the specimens listed in Table S1?

L362 – This has been changing through time so that I am not entirely sure about the latest way to cite properly the company for the Avizo software. However, to the best of my knowledge it should go as “FEI Visualization Sciences Group, Houston”

L387 – While I understand the will of the authors to summarize this information with the references they provide, I think it could be important to give (maybe in a table or as the authors prefer) the time of divergence of the fossil taxa included here in the analysis. It is true that they are already giving access to the actual phylogeny they use, but it is quite impractical having to extract node age information from the tree in R.

L402 – The authors report here that they have relied on Pagel’s lambda for estimating the phylogenetic signal. I was wondering if they have also considered computing Blomberg’s K as well, since the latter could provide information about the type of phylogenetic signal also (i.e., homoplasy, BM, stabilizing selection).

L403-405 – Did the authors computed also the cross-validated bgPCA? This praxis introduced by Cardini et al. (2020) allows to ascertain/exclude the presence of spurious grouping in the data.

L409 – Similar to the above, did the authors used cross-validated CV scores or not? Could they please specify?

Table S1 – Could the authors provide the references from which they have extracted the age for the considered taxa? This could be particularly relevant for those whose age is still controversial.

Figures – In several figures the authors refer to the “shadows” of the groups. I would like to suggest them a possible change with “silhouette”, which is more appropriate in my opinion.

Figure 1 - Here and in the other evolutionary plots (Figs. S1, S2) I am not sure that whether it could be viable due to space constraints in the figure, but it might be interesting to provide the morphology for key fossils that could help the reader to understand the direction of shape change through time, particularly due to the absence of extreme conformations for the CV axes.

Figure S5 – By looking at the picture, it seems that the authors forgot the numbers of the fixed landmarks

Figure S5 – “23 apex of the vestibular aqueduct” could the authors provide more information? Is it at the base of the “fan” formed by the aperture of the aqueduct?

Figure S5 – “15 curves containing semi-landmarks (SM) in yellow have been placed using the landmarks.” What the authors mean with “using the landmarks”? Could they clarify?

Reviewer #2:

Remarks to the Author:
Hidden in the deep
Mennecart et al.
reviewed by M. Clauss, Zurich

This work describes an analysis of the shape of the inner ear of ruminants, and links shapes to phylogenetic groups and past environmental changes.
The results appear to conform to expectations. The methodology, as far as I can judge, appears sound.

I have the following main comments, but also ask the authors to see the attached pdf file into which I made specific comments (e.g., on some citations, wording, some confusion in explaining phylogenetic position assignment in the methods etc.). Please also address those.

The graphics chosen to display the results are in my view not suitable. In theory, they should be 3-dimensional, but the perspective changes so that the uppermost layers (Pleistocene) is completely 2-dimensional. It is not possible to judge, from the graphics, the relative positions of the dots in the upper strata. I have no easy solution, but would think that it would be better to have all strata 3-dimensional, with "empty space" between strata (as it is, one cannot even know for some dots to which stratum they belong).

I would prefer a clear narrative of the different-level analyses, both in terms of the results they yield and the rationale why they were done.

1. PCA indicates to me (based on Fig. S1) that stem ruminantia, stem pecora, giraffids and cervids are all relatively similar, but that both bovids and tragulids are different (tragulids seem to be the furthest away from the stem ruminantia in Fig. S1).

2. bg-PCA Fig. S2 seems to indicate the same thing, you somehow say the same thing in the text but the fact that tragulids are quite different is somewhat not emphasized in my view.

3. CVA - now the tragulids seem to be like the stem ruminantia, and all others seem to be derived. I do not understand what these differences between the analyses mean, or why they occur.

I have the feeling that your narrative does not outline the rationale of the analysis (PCA -> bg-PCA -> CVA) and does not explain esp. the "change" in the position of the tragulids.

In general, while I understand that one cannot explain all methods in detail, the rationale of doing the different analyses should be explained. Being not familiar with the methods myself, I understand how one can use the PCA and also test for phylogenetic patterns, but I do not understand why the addition of phylogenetic information to the data (in bg-PCA and CVA, if I understood correctly) leads to the shifts in patterns mentioned above.

When describing the methods, please use the same words for all three analyses. E.g. if one analysis "does not standardize the within-group variance" - then I expect to read such a statement with the same words for the next analysis (i.e., it either also does not standardize the within-group variance, or it does - and it would be good to state what that means in terms of how the results can be interpreted).

I wondered - how do the results produced by using the inner ear shape differ from results just using species in the tree for the evolutionary rate calculations (I think one can do that just from the phylogeny), or from results based just on body size estimates. I trust your result as such, but I wonder whether it is really exceptional because of the BL shape, or could basically 'any' characteristic lead to the result (because it mainly depends on the species' tree?). It would be useful to contrast your BL results with others I suggested above, an evolutionary rate estimate based on the tree, and one based on body mass. I am aware this might be a huge amount of work and maybe there are good reasons not to do so. But without this, readers might not be convinced that doing all this work - BL scanning etc. - actually provides additional value to just using phylogenetic trees or much simpler characteristics.

sincerely marcus clauss

Reviewer #3:
Remarks to the Author:

This manuscript is based on the truly innovative, extremely well-executed idea of using the best-preserving bone of the mammalian skull – the petrosal and the inner ear canals it contains - to infer how this large and important mammalian clade diversified. Using excellent analyses of inner ear 3D morphology, the authors show that the diversification of ruminants into their main clades can be traced through time, and the data also provide convincing evidence that the diversification of these species can be associated with the evolution of inner ear shape, in terms of phylogenetic signal, radiation event timing, and evolutionary “speed” at different moments in geological history.

I hope that the manuscript will be published in Nature Communications. It deals with an iconic, well-known radiation of mammals, provides new and easily communicable information on this group, and has a clear narrative that will appeal to the readership of the journal. At the same time, the dataset is truly novel, and the methodology is highly rigorous. However, I have a few suggestions for revision related to the framing of the research (which currently undersells the importance of the manuscript), code functionality, data availability, and minor queries.

Background and Introduction

The text is a little rough in places – nothing serious, but the authors might consider reducing the jargon so it is more readable to non-specialists. Some sentences are quite long as well.

I believe the authors should emphasize more that it is nearly impossible to get an exhaustive deep-time dataset on the evolution for a group of mammals, unless we use petrosals (it is mentioned but not highlighted). Such a great idea.

I also like the premise that the inner ear has not been as functionally tractable as people have expected it to be, and that it probably has a lot of “drift” information in it. There are too many studies trying – and failing – to find some locomotor signal. I expect that some might disagree with this position but I am excited to see it published so we can cite it. I would make much more of this point because it is a core premise of the whole paper and a key innovation in how we use inner ear data!

Methods

The methods section is deceptively simple – a lot of thought effort went into the collection of the sample. The dataset is magnificent (and should be made publicly available for that reason – see below!). The analyses are to the point but not unnecessarily diverse - it is clear that the authors did not just throw every analysis available at the dataset.

The data and result documentation is very well organised and the analyses look great. It is great that the code is released and so transparent. However, I think there are a few parts of the workflow either hard to follow or possibly missing. I opened a couple of R files (the “Procedure_306” and “Procedure_191” files) and could not run the code because I couldn’t find the slider files for the curves and the source file for the subsampling. Some csv files (e.g. for `speclist.csv` in the `procedure_306` file) were also missing. It might also be good to have a brief explanation of what code is for what. It might be worth putting all of the code into github as well - this is not essential but it would give the authors better recognition and would make it easier for their code to be cited.

The implementation of RRPhylo for tracing evolutionary rate shifts is a great example of the use of this package. I know that its use for 3D analysis isn't universally liked in the GMM community, but I think it makes the case well. It might be worth briefly noting how the rates results are consistent with other analyses (e.g. CVA/PCA/Disparity results).

Because this paper is in the awkward Nature Communications format of Intro-Results/Discussion - Methods, it is hard to follow the methods section to first understand what the authors did. In particular, there is no rationale given for what question or hypothesis the authors addressed when employing a particular analysis. This an easy fix by adding a brief justification for each of the analyses (e.g. "To assess our expectation that there should be strong phylogenetic signal in our data, we...").

The data availability statement is currently still blank and marked in yellow in the ms. I would encourage the authors to make their 3D data available on an online platform like MorphoSource (gold standard and it is possible to receive a doi for specimens) or at least Figshare. If the authors are reluctant to freely share them immediately, they could put an embargo on them or at least make them downloadable upon request. If some data cannot be shared because of museum policy, it would still be good to share as much as possible.

Results/Discussion

The Results/Discussion part articulates the results clearly and integrates them with credible conclusions, although I am not very familiar with the clade. My main question would be how much bias in disparity, diversification rates, and evolutionary rates there might be because some clades are much less speciose than others at various times – there seems to be a danger of confounding species diversity in the sample with these metrics. For example, the greatest disparity among Moschidae and Tragulidae along CV1 and 2 is also when there were most contemporaneous species, while similarly speciose times for Giraffomorpha and Bovidae seem to also relate to their highest disparity. This could be discussed by pointing out examples where species numbers are clearly not an issue. Like in the Giraffomorpha, where evolutionary rates drop despite the high numbers of species after the late Miocene.

The figures are all just beautiful and convey the information very well. They are also backed up with exhaustive supplementary information. I found Fig. 1 a little hard to interpret because we are looking at the CV1/CV2 axes in a 3D arrangement and it is really hard to know how the points related to each other on a scatterplot CV1 vs CV2. Thus, it seems like Giraffomorphs and Cervids are not distinguishable in morphospace but I wonder if this is an artifact of the perspective? Would it be worth running a procrustes analysis with these clades as factors to ask if the differences picked up by the CVA are significant?

Some of the supplementary figures do not seem to be referred to in the text.

I am happy to be contacted if I can help at all.

Vera Weisbecker

Dear referee,

We have been following your comments and corrections. Thank you very much for your help improving our manuscript.

Bastien Mennecart

REVIEWER COMMENTS

Reviewer #1 (Remarks to the Author):

The manuscript titled “Hidden in the deep: ruminant inner ear sensory organs record 35 million years of major historical events” relies on shape changes occurring in the ruminant inner ear bony labyrinth for tracing the evolutionary history of this successful group of artiodactyls. To do so, the authors are relying on a previously described 3DGM approach (also slightly updated), which is applied to a very large sample of extant and fossil ruminants. The shape data is then analyzed by means of recently developed methods, which have not been applied to BL morphology before. Based on their results, the authors observe that changes in BL evolutionary rates occur in correspondence of major paleoecological events and conclude that BL morphology is well suited as a proxy for studying adaptive radiations. As a whole, this contribution appears well conceived and very comprehensive (especially sample-wise), with adequate descriptions of the methods and a massive amount of additional data, with the latter that is very useful for further supporting the conclusions reached, as well as to ensure the repeatability of the analyses. Thanks to the multidisciplinary approach employed, the novelty of the methods and the far-reaching conclusions that could be drawn, the results here presented constitute a remarkable advance in multiple fields related with evolutionary studies, and will be of interest to a large audience of both specialized and unspecialized readers.

The numerous positive aspects detailed above notwithstanding, there are a few points that would like to bring to the attention of the authors and ask them to consider.

Overall, a very interesting contribution
Alessandro Urciuoli

General comments:

- The overall format of the paper is rather unusual in my opinion. I understand that this is mostly an editorial matter, but the authors do not seem to have followed the format suggested by the journal neither in the abstract (>150 words long) nor in the sections in which the paper should be subdivided. Accordingly, I would suggest the authors to follow a more linear and “standard” structure, which should possibly include Introduction, Results and Discussion sections.

The manuscript has been restructured in a more classical way.

- Regardless of whether the authors would re-structure their paper, I would like to suggest them to provide a broader background to their study in which they should expand the various topics that they are already providing in the current draft e.g., why climate change is an important evolutionary

factor? What has been done already in bony labyrinth studies? What is the state of the art about ruminant phylogeny and evolution? Which are the radiation events the authors are willing to trace? I think that this might be most useful for the readers, especially unspecialized ones, to obtain insights about why the authors are relying on ruminants and BL morphology to test their evolutionary hypotheses.

The abstract and introduction has been entirely rewritten

- Similar to the above, the authors in the current version are almost always directly discussing the results without previously present them to the reader. While the huge amount of supplementary data allows a correct interpretation, this requires a considerable amount of effort for the reader, which has to constantly switch between the main text and the supplementary data. This particularly applies to the different morphospaces here presented (PCA, bgPCA and CVA), which lack a proper description of the distribution of the taxa and of the patterns of shape variation observed along the considered axes (PC1-2, bgPC1-2, CV1-2). Since the current online content of the manuscript amounts to ~2000 words (with 5000 being the maximum), the authors should be able to include additional descriptions of their results and treat them separately in the discussion.

A result part has been added

Specific comments

L49 – “historical parameters” I am not sure about the meaning the authors are intending with “historical”. Please clarify

The abstract has been entirely rewritten

L52 – “of the inner ear of the iconic ruminants” Please remove “the”

The abstract has been entirely rewritten

L60 – “fostered the development of heterogeneous environments promoting diversification and evolutionary rates in these clades” Possibly the authors omitted “fast” before “evolutionary rates”? Also, could the authors specify which are the clades they are referring to? Both deer and bovinds, ruminants as a whole, or other ruminant clades?

The abstract has been entirely rewritten

L61-63 – I understand that the authors are willing one of the reasons for which they reached the conclusions mentioned in the period above. However, the current wording does not provide an adequate connection between the two periods. Please consider rephrasing.

The abstract has been entirely rewritten

L74 – “of modern and fossil mammals” is the previous statement restricted to mammals only? If it is the case, could the authors explain why and provide reference?

Changed for

The petrosal bone itself and the BL, mostly since the advent of CT-scanning techniques allowing assembling a virtual endocast, have helped to understand various aspects of mammalian evolution. Examples are the early steps of mammalian evolution in the Mesozoic²², the mammalian phylogeny reconstructed using the wealth of morphological characters they can yield^{20,23,24}, and the evolution and dispersal of modern humans from Africa²¹

L75 – “An emerging anatomical object in evolutionary research that relates to the relationships (i.e. phylogeny) of animals”. The wording of this sentence is somewhat unclear, even if overall correct. I would suggest to the authors to consider the wording “An emerging anatomical object in evolutionary research thought to accurately reflect the kinship (i.e., phylogeny) between different species”

We have followed your proposition

The bony labyrinth (BL) is an emerging anatomical structure in evolutionary research preliminary proved to accurately reflect the kinship (i.e., phylogeny) between species —and hence their evolution over time^{20,21}.

L79 – “outstanding analyses”. This is unclear to me. Could the authors clarify what are “outstanding analyses” in their view? Were they referring to recently developed methods?

It has been changed for

Yet, data show that BL morphology may only partly depend on ecological and physiological parameters such as habitat type^{25,26}, or locomotion capabilities^{27,28}.

L80 – “few” The use of this adjective is not correct. Please consider rephrasing.

The sentence has been deleted

L80-81 – “Then this structure may neutrally evolve like in genetic drift” Might authors expand on this concept? I understand that they are willing to highlight that the absence of hard functional constraints allowed the BL morphology to evolve neutrally. However, the wording appears slightly unclear and the connection with the previous sentence is not very easy to be followed.

The concept appears now latter and is explained in the light of other publications

Predominantly, the morphological signal recorded in the BL is strongly related to phylogeny^{20,29} and is considered to reflect neutral evolution³⁰. The low selection pressure on this morphological structure may be due to its important developmental constraints²⁰.

L81-86 – Similarly to what happens with the previous sentence, this paragraph changes subject in a quite abrupt manner and does not connect appropriately with the previous one.

It has been rewritten

L91 – CT or microCT? The authors are previously referring to the latter.

Done

L94 – “general” Generalist? Unspecialized? Primitive? General is too vague in my opinion. Please clarify.

Changed for an overall

L102-103 – Either “Pronghorn BL morphologies are the closest to those of the stem pecorans, while those of bovids are more derived,” or “Pronghorn BL morphology is the closest to that of the stem pecorans, while that of bovids is more derived”. Also, could the author provide information about why the observed morphologies are primitive or derived?

Changed for Pronghorn BL morphologies are the closest to **those** of the stem pecorans

L161 – “limited” did the authors mean “limited to”?

Indeed

L167 – “cooling reducing” possibly replace with “, which reduced”

done

L176-178 – I would suggest the authors to connect the two sentences with “where at least 19 extant species evolved locally and currently represent at least”

done

L178-182 – Very long sentence, consider splitting at least with commas or into separate sentences.

we have added commas

L184 – “stayed” and “little”, both words are not fitting entirely well in my opinion to the sentence and I would like to suggest the authors to consider replacing them with “remained” and “poorly”, respectively.

done

L184-185 – “and mostly inhabited rather open and similar habitats throughout their entire history” could this possibly be the cause of the low diversification? If so, the authors should be more explicit about it.

We have completed this part.

L185-188 – The authors here provide possible explanations about the very distinct morphology of Antilocapra relative to the other representant of the clade. However, and as specified above for the derived/primitive condition of bovids and pronghorns, in which terms is Antilocapra derived? The reader can indeed spot some differences between Fig. S3g and h (the authors could also directly refer to these frames in Fig. S3 for clarity) yet no general description of the bony labyrinth is provided so it seems somewhat unclear to me. In my opinion, the authors should provide a brief resume about which are the primitive features found in fossil antilocaprids, as well as the derived ones of extant antilocapra. In addition to the above, does this statement hold true for more recent species such as Stockoceros and Capromeryx? I understand that the space constraints for the already very large Fig.

S3 does not allow to add them, but the absence of the latter two taxa in the figure does not allow to ascertain this.

The manuscript has been restructured and more details on the morphology has been provided.

a derived BL morphology for *Antilocapra* in comparison to those of fossil antilocaprids (Fig.1 G-H, e.g., shape of the vestibular and cochlear aqueducts) that may have driven the acceleration in rates of morphological evolution.

L200-208 – The authors are here delineating their conclusions, but the narration appears jumpy as the authors are changing topic in a somewhat abrupt way. I agree with them about the similar pattern observed in the evolution of body mass and BL shape. However, while reading this paragraph it is not clear, at first glance, why the authors are bringing up this here. The author could try to give a brief explanation (even better if this is done in the introduction) about why the BM evolutionary rates are a good proxy for inferring climatic fluctuations.

This section has been rewritten

The evolution of BL morphology in extant and extinct ruminants confirm that climate and tectonics (extrinsic factors), as well as biological traits and ecological opportunities (intrinsic factors) over long time scales have all contributed in triggering the global morphological diversification and diversity. BL shape can reflect, but only at best in extreme cases, functional requirements under strong selection pressure. However, a growing body of evidence shows that it is more strongly correlated to phylogeny^{20,29}. Neutral evolutionary processes seem to be driving its morphological evolution, which is visible at the microevolutionary level, since fine scale BL shape differences in populations of closely related species are significant^{77,105}. Our contrasting results on evolutionary rate shifts in various groups show striking similarities to molecular evolutionary rates. They confirm that BL shape is a powerful tool for investigating the complex links between morphology and evolution, and their forcing factors over geological timescales. These new results complement those coming from the analysis of molecular data derived from extant taxa. Combining both data sources is likely to yield a clear picture on the evolutionary history of mammals and, specifically, on the forcing factors shaping their diversity and evolution.

L203-205 – In my opinion the authors should consider rephrasing this period to clarify what is the evidence they found in their data and how this reflects the changes occurring in paleoecological factors. At present, it appears to be the other way around.

This section has been rewritten

L203-208 – The periods found here in sequence do not connect adequately one to the other and are overall somewhat sketchy. Since these represent the take-home-message of the paper, I would like to suggest the authors to enhance their clarity.

This section has been rewritten

L359 – The authors refer here to the range of voxel size for the microCT data. Could the authors provide pixel resolution and facility for each of the specimens listed in Table S1?

L362 – This has been changing through time so that I am not entirely sure about the latest way to cite properly the company for the Avizo software. However, to the best of my knowledge it should go as “FEI Visualization Sciences Group, Houston”

done

L387 – While I understand the will of the authors to summarize this information with the references they provide, I think it could be important to give (maybe in a table or as the authors prefer) the time of divergence of the fossil taxa included here in the analysis. It is true that they are already giving access to the actual phylogeny they use, but it is quite impractical having to extract node age information from the tree in R.

We have added

The nodes of the main clades are calibrated using the fossil record. The oldest known ruminant is *Archaeomeryx* from the middle Eocene of Asia, ca. 44 Ma⁵⁹. The origin of the crown Pecora is older than 37 Ma⁶⁰. The giraffomorph *Bedenomeryx* is known from 24 Ma⁶¹. The oldest moschid is ca. 18 Ma⁶², while the oldest bovid ca. 18.75 Ma. Node calibration within families is estimated based on several sources in the literature^{20,50,63}. The phylogenetic tree can be consulted in Fig. S3 and the nexus file is given in the Supplementary Data 1.

L402 – The authors report here that they have relied on Pagel’s lambda for estimating the phylogenetic signal. I was wondering if they have also considered computing Blomberg’s K as well, since the latter could provide information about the type of phylogenetic signal also (i.e., homoplasy, BM, stabilizing selection).

K mult results can be found in supplementary data 1. It is 0.36, so way different from Lambda.

We have found this explanation (<http://blog.phytools.org/2012/03/phylogenetic-signal-with-k-and.html>)

K & λ are not the same thing. λ measures the similarity of the covariances among species to the covariances expected under Brownian motion; whereas K might be more usefully thought of as a measure of the partitioning of variance. If $K > 1$ then variance tends to be among clades; while if $K < 1$ then variance is within clades (with BM as reference). The variance on K for a given process is quite large. I suspect if you did simulation you might find that $K = 0.54$ (particularly for a relatively small tree) was not different from BM.

So we changed for “The PCA polymorphospace seems to represent a phylomorphospace, as demonstrated by the test on the phylogenetic signal (permutation test $p < 0.001$). The dataset displays similar covariances among species to the covariances expected under Brownian motion (Pagel’s λ 0.83, $p < 0.001$). However, the variance is mostly within clade, not between clades (Blomberg’s K 0.36).”

L403-405 – Did the authors computed also the cross-validated bgPCA? This praxis introduced by Cardini et al. (2020) allows to ascertain/exclude the presence of spurious grouping in the data. cross validation have been performed and can be consulted in supplementary data1. We also mention the results.

L409 – Similar to the above, did the authors use cross-validated CV scores or not? Could they please specify?

cross validation have been performed and can be consulted in supplementary data1. We also mention the results.

Table S1 – Could the authors provide the references from which they have extracted the age for the considered taxa? This could be particularly relevant for those whose age is still controversial.

The ages correspond to the age of the localities where the fossil has been found, not the range of the species.

Figures – In several figures the authors refer to the “shadows” of the groups. I would like to suggest them a possible change with “silhouette”, which is more appropriate in my opinion.

done

Figure 1 - Here and in the other evolutionary plots (Figs. S1, S2) I am not sure that whether it could be viable due to space constraints in the figure, but it might be interesting to provide the morphology for key fossils that could help the reader to understand the direction of shape change through time, particularly due to the absence of extreme conformations for the CV axes.

Figure S3 (first plate) has been included into the MS. The specimens from the figure is now highlighted in the Figure 2 (ex-Figure 1).

Figure S5 – By looking at the picture, it seems that the authors forgot the numbers of the fixed landmarks

This has been fixed

Figure S5 – “23 apex of the vestibular aqueduct” could the authors provide more information? Is it at the base of the “fan” formed by the aperture of the aqueduct?

added at the base of the endolymphatic sac

Figure S5 – “15 curves containing semi-landmarks (SM) in yellow have been placed using the landmarks.” What the authors mean with “using the landmarks”? Could they clarify?

The semi-landmarks have been placed **between** the landmarks.

Reviewer #2 (Remarks to the Author):

Hidden in the deep

Mennecart et al.

reviewed by M. Clauss, Zurich

This work describes an analysis of the shape of the inner ear of ruminants, and links shapes to

phylogenetic groups and past environmental changes.

The results appear to conform to expectations. The methodology, as far as I can judge, appears sound.

I have the following main comments, but also ask the authors to see the attached pdf file into which I made specific comments (e.g., on some citations, wording, some confusion in explaining phylogenetic position assignment in the methods etc.). Please also address those.

Thank you. All your comments have been followed

The graphics chosen to display the results are in my view not suitable. In theory, they should be 3-dimensional, but the perspective changes so that the uppermost layers (Pleistocene) is completely 2-dimensional. It is not possible to judge, from the graphics, the relative positions of the dots in the upper strata. I have no easy solution, but would think that it would be better to have all strata 3-dimensional, with "empty space" between strata (as it is, one cannot even know for some dots to which stratum they belong).

All the 3D graphs are provided in the supplementary files using a copy past in R. In addition, we have provided an animated gif to show in 3D the graphical repartition of the points (supplementary data1).

I would prefer a clear narrative of the different-level analyses, both in terms of the results they yield and the rationale why they were done.

Here is the answer of reviewer 3: "has a clear narrative that will appeal to the readership of the journal". The comments of reviewer 2 goes in the other direction than reviewer 3 on these specific points. We change the MS to make it clearer and hope that reviewer2 will appreciate.

1. PCA indicates to me (based on Fig. S1) that stem ruminantia, stem pecora, giraffids and cervids are all relatively similar, but that both bovids and tragulids are different (tragulids seem to be the furthest away from the stem ruminantia in Fig. S1).

2. bg-PCA Fig. S2 seems to indicate the same thing, you somehow say the same thing in the text but the fact that tragulids are quite different is somewhat not emphasized in my view.

3. CVA - now the tragulids seem to be like the stem ruminantia, and all others seem to be derived. I do not understand what these differences between the analyses mean, or why they occur.

I have the feeling that your narrative does not outline the rationale of the analysis (PCA -> bg-PCA -> CVA) and does not explain esp. the "change" in the position of the tragulids.

In general, while I understand that one cannot explain all methods in detail, the rationale of doing the different analyses should be explained. Being not familiar with the methods myself, I understand how one can use the PCA and also test for phylogenetic patterns, but I do not understand why the addition of phylogenetic information to the data (in bg-PCA and CVA, if I understood correctly) leads to the shifts in patterns mentioned above.

The 3 methods are different kind of analyses. The PCA display the variation within the dataset. This is not a test by itself and you can not infer anything from it without an additional test. This is why we used other test bg-PCA and CVA to characterize what is morphologically typical here for a clade. So these are different kind of analyses for different kind of questions. Bg-PCA and CVA are explained in the MS .

"To characterize the morphological similarities within the different clades, a between-groups PCA (bg-PCA) and a Canonical Variates Analysis (CVA) were performed. bg-PCA and CVA provide complementary information^{71,72}.

A bg-PCA observes the variance between groups (here defined as Stem Ruminantia, Tragulidae, Stem Pecora, Antilocapridae, Giraffomorpha, Cervidae, Moschidae, and Bovidae) without standardizing the within-groups variance⁷¹. A standardization is performed when using the CVA. The CVA maximizes the separation of the between-groups means relative to the variation within the groups' ratio according to the specified chosen grouping variable⁷¹. The bg-PCA was computed using the function "groupPCA" and the CVA using the function "CVA", both in the R package *Morpho* v2.9⁶⁵. To test the performance of the classification model, both analyses were cross-validated using leaving-one-out cross-validation for the bg-PCA and Jackknife Cross-validation for the CVA."

We have added for the PCA "We performed a Principal Component Analysis (PCA) computed using the function "procSym" in the R package *Morpho* v2.9⁶⁵ to study the shape variation within the dataset in its natural scale."

Then when considering the broad evolution, this is the general shape variation that we are considering using PCA results. If we are considering the cladogenesis, we are interested on what is characteristic of a group and what groups is most closely related to another one using an a priori analysis.

When describing the methods, please use the same words for all three analyses. E.g. if one analysis "does not standardize the within-group variance" - then I expect to read such a statement with the same words for the next analysis (i.e., it either also does not standardize the within-group variance, or it does - and it would be good to state what that means in terms of how the results can be interpreted).

This has been changed for

"To characterize the morphological similarities within the different clades, a between-groups PCA (bg-PCA) and a Canonical Variates Analysis (CVA) were performed. bg-PCA and CVA provide complementary information^{71,72}. A bg-PCA observes the variance between groups (here defined as Stem Ruminantia, Tragulidae, Stem Pecora, Antilocapridae, Giraffomorpha, Cervidae, Moschidae, and Bovidae) without standardizing the within-groups variance⁷¹. A standardization is performed when using the CVA. The CVA maximizes the separation of the between-groups means relative to the variation within the groups' ratio according to the specified chosen grouping variable⁷¹. The bg-PCA was computed using the function "groupPCA" and the CVA using the function "CVA", both in the R package *Morpho* v2.9⁶⁵. To test the performance of the classification model, both analyses were cross-validated using leaving-one-out cross-validation for the bg-PCA and Jackknife Cross-validation for the CVA."

I wondered - how do the results produced by using the inner ear shape differ from results just using species in the tree for the evolutionary rate calculations (I think one can do that just from the phylogeny), or from results based just on body size estimates. I trust your result as such, but I wonder whether it is really exceptional because of the BL shape, or could basically 'any' characteristic lead to the result (because it mainly depends on the species' tree?). It would be useful to contrast your BL results with others I suggested above, an evolutionary rate estimate based on the tree, and one based on body mass. I am aware this might be a huge amount of work and maybe there are good reasons not to do so. But without this, readers might not be convinced that doing all this work - BL scanning etc. - actually provides additional value to just using phylogenetic trees or much simpler characteristics.

If you explore the supplementary data, you will see that we have done all the analyses on the centroid size. As already demonstrated by Costeur et al. 2019 there is a direct correlation between the BL size and the body mass. You will notice that the evolutionary rates of the centroid size are different from what is produced with the BL morphology.

Loïc Costeur, Bastien Mennecart, Bert Müller, Georg Schulz, "Observations on the scaling relationship between bony labyrinth, skull size and body mass in ruminants," Proc. SPIE 11113, Developments in X-Ray Tomography XII, 1111313 (24 September 2019); doi: 10.1117/12.2530702

We have then added this

Similar analyses have also been performed based on the BL centroid size. BL size has been proved to be a good proxy for body mass estimation in ruminants⁷⁶ and has been used here to explore size evolution and diversification in ruminants. The results of these analyses are given in Supplementary Data 1. We find that they differ significantly from results based on BL morphological evolution (evolutionary rate and significant shifts, as well as large-scale correlation with environmental factors) which indicates that the topology of the tree and the number of considered taxa are not the main factor driving our results.

the topmost rows of red and dark blue dots that are aligned "like on a string" is strange - is this true, data that indicates this degree of similarity by one of the axes?

Indeed they are in line, they are all from specimens of today

I do not see the acceleration in bovids in Fig. 2 - what do you mean here?

The branches are green, not blue. This means there is an acceleration of the evolution as defined in the figure caption.

I tried to understand why 37 was cited for "outpassing other potential competitors" but could find not indication for that in paper 37. Either correct citation or explain with a few words in the main text why this citation supports this claim

This has been deleted

what is this (extinct) diversification hotspot in the bovids?

This is the late middle Miocene "Gazelles" from Europe, but this is not a significant change in the evolutionary rates.

don't the method of "evolutionary rate calculation" account for the number of species for which measurements are available? If not, then any more speciose clade should be the one with the higher evolutionary rate, no?

Indeed, the number of species and the branch length is considered when doing a rate. Morphological evolutionary rates and taxonomical diversification rates are not the same things. For example the Tragulidae are speciose with a low evolutionary rate. The insular species are little diversified with a high evolutionary rate.

As mentioned before, when using other data (as the bodymass/centroid size), the results may be different:

Similar analyses have also been performed based on the BL centroid size. BL size has been proved to be a good proxy for body mass estimation in ruminants⁷⁶ and has been used here to explore size evolution and diversification in ruminants. The results of these analyses are given in Supplementary Data 1. We find that they differ significantly from results based on BL morphological evolution (evolutionary rate and significant shifts, as well as large-scale correlation with environmental factors) which indicates that the topology of the tree and the number of considered taxa are not the main factor driving our results.

I apologize for not being familiar with the methods at all - but in all this, is there some kind of control for the number of species available in a clade? If there are more species in a clade, would one not expect some slight increase in BL shape variety just because of that fact?

The diversity may increase, but the rate not necessary since it is divided by the cumulative branch length.

in reference 56, in my understanding, the extant giraffe is included in the giraffomorphs, as you write below in the pink sentence ... but if the giraffomorpha are a sister clade to the Bovoidea (orange sentence), how can the Bovoidea be composed of Giraffidae (blue sentence)?

Indeed this is a mistake when restructuring the sentence. the Bovoidea composed of the Moschidae and Bovidae

I do not understand how you apply a phylogenetic framework when creating a mean shape for a species?

When several specimens were available for one species, we created a mean shape for the species using the function "*mshape*" of the package *geomorph*^{60,61} for the analysis that are using a phylogenetical framework.

Please make the layout of the supplementary information more intuitive.

Figures should have their legend BELOW them (as in the main text), and each Figure should be on a new page where the figure is on top and the legend below.

Done

Table S1 – Could the authors provide the references from which they have extracted the age for the considered taxa? This could be particularly relevant for those whose age is still controversial.

This is the age of the locality, not of the species (range). This has been precise in the Table S1.

Figure S4. Evolutionary rates of all the ruminant BL morphology based on PC scores compared with the global temperature curve¹³ (grey dots). The red dash line is the evolutionary rates of the stem and crown Pecora, the dark blue dash line is the evolutionary rates of the stem Ruminantia and Tragulidae. Red complete line with associated silhouette indicate clades included in Pecora with the biostratigraphic range of the stem and crown Pecora and dark blue complete line with associated silhouette indicate clades included in stem Ruminantia and Tragulidae with their biostratigraphic range. Important biotic (GABI = Great American Biotic Interchange) and abiotic (TEE = Terminal Eocene Event, Mi1 = first Miocene glaciation, mMCO = middle Miocene Climatic Optimum). Events in blue are related to global cooling, while the event in yellow is linked to global warming. Methodology and statistical results produced by the R package RPanda⁴⁹ are provided in “Material and Methods” and Supplementary Data 1, 2. Silhouette of the families modified from²⁷.

sincerely marcus clauss

Reviewer #3 (Remarks to the Author):

This manuscript is based on the truly innovative, extremely well-executed idea of using the best-preserving bone of the mammalian skull – the petrosal and the inner ear canals it contains - to infer how this large and important mammalian clade diversified. Using excellent analyses of inner ear 3D morphology, the authors show that the diversification of ruminants into their main clades can be traced through time, and the data also provide convincing evidence that the diversification of these species can be associated with the evolution of inner ear shape, in terms of phylogenetic signal, radiation event timing, and evolutionary “speed” at different moments in geological history.

I hope that the manuscript will be published in Nature Communications. It deals with an iconic, well-known radiation of mammals, provides new and easily communicable information on this group, and has a clear narrative that will appeal to the readership of the journal. At the same time, the dataset is truly novel, and the methodology is highly rigorous. However, I have a few suggestions for revision related to the framing of the research (which currently undersells the importance of the manuscript), code functionality, data availability, and minor queries.

Thank you very much.

Background and Introduction

The text is a little rough in places – nothing serious, but the authors might consider reducing the jargon so it is more readable to non-specialists. Some sentences are quite long as well.

We have restructured and modified the text to make it easier.

I believe the authors should emphasize more that it is nearly impossible to get an exhaustive deep-

time dataset on the evolution for a group of mammals, unless we use petrosals (it is mentioned but not highlighted). Such a great idea.

Thank you very much.

I also like the premise that the inner ear has not been as functionally tractable as people have expected it to be, and that it probably has a lot of “drift” information in it. There are too many studies trying – and failing – to find some locomotor signal. I expect that some might disagree with this position but I am excited to see it published so we can cite it. I would make much more of this point because it is a core premise of the whole paper and a key innovation in how we use inner ear data!

Thank you very much.

Methods

The methods section is deceptively simple – a lot of thought effort went into the collection of the sample. The dataset is magnificent (and should be made publicly available for that reason – see below!). The analyses are to the point but not unnecessarily diverse - it is clear that the authors did not just throw every analysis available at the dataset.

Thank you very much.

The data and result documentation is very well organised and the analyses look great. It is great that the code is released and so transparent. However, I think there are a few parts of the workflow either hard to follow or possibly missing. I opened a couple of R files (the “Procedure_306” and “Procedure_191” files) and could not run the code because I couldn’t find the slider files for the curves and the source file for the subsampling. Some csv files (e.g. for speclist.csv in the procedure_306 file) were also missing. It might also be good to have a brief explanation of what code is for what. It might be worth putting all of the code into github as well - this is not essential but it would give the authors better recognition and would make it easier for their code to be cited.

The text has restructured and the R code has been cleaned.

The implementation of RRPhylo for tracing evolutionary rate shifts is a great example of the use of this package. I know that its use for 3D analysis isn’t universally liked in the GMM community, but I think it makes the case well. It might be worth briefly noting how the rates results are consistent with other analyses (e.g. CVA/PCA/Disparity results).

Thank you very much.

Because this paper is in the awkward Nature Communications format of Intro-Results/Discussion - Methods, it is hard to follow the methods section to first understand what the authors did. In particular, there is no rationale given for what question or hypothesis the authors addressed when

employing a particular analysis. This an easy fix by adding a brief justification for each of the analyses (e.g. "To assess our expectation that there should be strong phylogenetic signal in our data, we...").

We have reordered the MS. The format for the first submission was the Nature one.

We have added this

We performed a Principal Component Analysis (PCA) computed using the function "procSym" in the R package *Morpho* v2.9⁶⁵ to study the shape variation within the dataset in its natural scale. In order to assess our expectation that there should be a strong phylogenetic signal in our data, we performed a permutation test (randomized rounds: 10.000⁶⁶) based on the phylogenetic tree. Klingenberg and Gidaszewski⁶⁶ defined that "The empirical p-value for the test is the proportion of permuted data sets in which the sum of squared changes is shorter or equal to the value obtained for the original data." –Marriott⁶⁷ and Edgington⁶⁸ suggested that 1.000 permutations are a reasonable minimum for a test at 5% level of significance, while 5.000 are a reasonable minimum at the 1% level⁶⁹. Moreover, the Pagel's λ and Bloomberg's K phylogenetic signal values have been calculated using the function "phylosig" in the R package *phytools* v1.0.3⁷⁰. All supporting data are given in the Supplementary Data 1, 2. To characterize the morphological similarities within the different clades, a between-groups PCA (bg-PCA) and a Canonical Variates Analysis (CVA) were performed. bg-PCA and CVA provide complementary information^{71,72}. A bg-PCA observes the variance between groups (here defined as Stem Ruminantia, Tragulidae, Stem Pecora, Antilocapridae, Giraffomorpha, Cervidae, Moschidae, and Bovidae) without standardizing the within-groups variance⁷¹. A standardization is performed when using the CVA. The CVA maximizes the separation of the between-groups means relative to the variation within the groups' ratio according to the specified chosen grouping variable⁷¹. The bg-PCA was computed using the function "groupPCA" and the CVA using the function "CVA", both in the R package *Morpho* v2.9⁶⁵. To test the performance of the classification model, both analyses were cross-validated using leaving-one-out cross-validation for the bg-PCA and Jackknife Cross-validation for the CVA.

The data availability statement is currently still blank and marked in yellow in the ms. I would encourage the authors to make their 3D data available on an online platform like MorphoSource (gold standard and it is possible to receive a doi for specimens) or at least Figshare. If the authors are reluctant to freely share them immediately, they could put an embargo on them or at least make them downloadable upon request. If some data cannot be shared because of museum policy, it would still be good to share as much as possible.

All the landmarks of all the specimens are associated to the manuscript. We are also willing to provide access to the 3D reconstruction. We have already provided open access to some of the studied BL and we will continue to do so:

Mennecart B., Métais G., Tissier J., Rössner G.E., & Costeur L. (2021). 3D models related to the publication: Reassessment of the enigmatic ruminant Miocene genus *Amphimoschus* Bourgeois, 1873 (Mammalia, Artiodactyla, Ruminantia, Pecora). *MorphoMuseum* 7, e131. DOI 10.18563/journal.m3.131

Mennecart B., Perthuis de Ad., & Costeur L. (2018). 3D models related to the publication: The first French tragulid skull (Mammalia, Ruminantia, Tragulidae) and associated tragulid remains from the Middle Miocene of Contres (Loir-et-Cher, France). *MorphoMuseum*. DOI 10.18563/journal.m3.3.3.e4.

Aiglstorfer M., Costeur L., **Mennecart B.**, & Heizmann E.P.J. (2107). *Micromeryx? eiselei* - a new moschid species from Steinheim am Albuch, Germany, and the first comprehensive description of moschid cranial material from the Miocene of Central Europe. *MorphoMuseum* 3(4)-e4. DOI 10.18563/03.3.4.e4.

Costeur L. & **Mennecart B.** (2016). 3D models related to the publication: Prenatal growth stages show the development of the ruminant bony labyrinth and petrosal bone. *MorphoMuseum*. 2(2)-e3. DOI 10.18563/m3.2.2.e3.

Mennecart B. & Costeur L. (2016b). 3D models related to the publication: *A Dorcatherium* (Mammalia, Ruminantia, Middle Miocene) petrosal bone and the tragulid ear region. *MorphoMuseum*. 2 (1)-e2. DOI 10.18563/m3.2.1.e2.

However, as mentioned in Costeur and Mennecart (2019), we are right now writing a catalogue of the ear region in ruminants including descriptions of all the species from this article. We would like to release in open access a maximum of 3D reconstructions with this article.

Costeur L. & Mennecart B. (2019) Building a catalogue of the ear region for ruminants. *Journal of morphology* 280, S101-S102.

Results/Discussion

The Results/Discussion part articulates the results clearly and integrates them with credible conclusions, although I am not very familiar with the clade. My main question would be how much bias in disparity, diversification rates, and evolutionary rates there might be because some clades are much less speciose than others at various times – there seems to be a danger of confounding species diversity in the sample with these metrics. For example, the greatest disparity among Moschidae and Tragulidae along CV1 and 2 is also when there were most contemporaneous species, while similarly speciose times for Giraffomorpha and Bovidae seem to also relate to their highest disparity. This could be discussed by pointing out examples where species numbers are clearly not an issue. Like in the Giraffomorpha, where evolutionary rates drop despite the high numbers of species after the late Miocene.

Thank you, indeed this is also the case when considering the Antilocapridae. Only one species remains today but we observe a huge increase in its evolutionary rates. Similarly, when considering other kind of data (BL centroid size that is a proxy for body mass) we obtain different kind of results. We tried to highlight it more in the MS.

Similar analyses have also been performed based on the BL centroid size. BL size has been proved to be a good proxy for body mass estimation in ruminants⁷⁶ and has been used here to explore size evolution and diversification in ruminants. The results of these analyses are given in Supplementary Data 1. We find that they differ significantly from results based on BL morphological evolution (evolutionary rate and significant shifts, as well as large-scale correlation with environmental factors) which indicates that the topology of the tree and the number of considered taxa are not the main factor driving our results.

The figures are all just beautiful and convey the information very well. They are also backed up with exhaustive supplementary information. I found Fig. 1 a little hard to interpret because we are looking at the CV1/CV2 axes in a 3D arrangement and it is really hard to know how the points related to each other on a scatterplot CV1 vs CV2. Thus, it seems like Giraffomorphs and Cervids are not distinguishable in morphospace but I wonder if this is an artifact of the perspective? Would it be worth running a procD.gls analysis with these clades as factors to ask if the differences picked up by the CVA are significant?

Cross validation has been performed for the bg-PCA and CVA. Moreover, the 3D data of the plots are now available as an animated GIF in supplementary data.

Some of the supplementary figures do not seem to be referred to in the text.

We corrected it.

I am happy to be contacted if I can help at all.

Vera Weisbecker

Reviewers' Comments:

Reviewer #1:

Remarks to the Author:

The revised version of the manuscript by Mennecart and coauthors titled "Hidden in the deep: ruminant inner ear sensory organs record 35 million years of major historical events" shows a great improvement in its clarity and in the way the results are presented to the reader. Particularly, the structure is now much easier to be followed and guides much better the reader through the very interesting conclusions reached by the authors. I am more than satisfied by the way the authors dealt with the suggestions provided and implemented them into the new version of main text. I think that the manuscript will be perfectly suitable for publication as soon as the authors fix a few quite minor points (see below).

Overall, I consider that this manuscript will be a key contribution in inner ear studies, especially as it makes clear that some of the most-commonly accepted paradigms (e.g., the relationship between canal shape and locomotion) should be considered more carefully than what previously suggested. A job well done, congratulations!

Alessandro Urciuoli

As a general minor comment regards the use of the Supplementary Data 1. The authors often refer to it throughout the text. However, the compressed folder is actually quite large and includes a lot of very valuable information. Would there be a different manner to cite this information that could more clearly state to which of the files the authors are referring to each time?

The rest of the suggestions can be found on the merged PDF with tracked changes with the following line references:

-L86-87: The sentence starting with "Body mass..." seems to be misplaced between two others that deal with the issues related with the fragmentary nature of the fossil record. Possibly the authors could move it right after the sentence ending at L85 with "...evolutionary rates"?

-L119: I think that "the" is missing in between the words "to evolution"

-L194: even if I understand that "cochlea" is used here as an adjective, the wording results somewhat difficult to be followed. As an alternative, the authors could use "first cochlear turn thickness".

-L216: I would suggest the authors to change "in" for "of"

-L262: when answering to a point I raised for the previous version of the manuscript, the authors are stating that "the PCA polymorphospace seems to represent a phylomorphospace". What do the authors mean? A phylomorphospace is just obtained by projecting a phylogenetic tree onto a morphospace, regardless of the observed distribution of the tips in the morphospace. To my understanding, the permutation test ($p < 0.001$) that they have performed suggest that the average amount of shape change along the branches of the tree is relatively small due to the presence of phylogenetic signal (based on the definition of the test by Klingenberg and Gidaszewski).

-L266: As per its design, the PCA accumulates the greatest amount of variance on the first PC. I would suggest the authors to directly refer to the amount of variance explained by this PC.

-L276: Missing word capitalization within the parentheses.

-L281: I would suggest the authors to change "anterior" for "anteriorly" and to add the article "the" right before "posterior"

-L291: the authors here refer to the fact that Bovidae is the most variable group among those considered. Could the authors provide a very brief recapitulation sentence stating which are the extremes of this range of variation (e.g., "including species with both very short and very long cochlea")? If the authors consider that this might be redundant, please ignore this comment.

-L313: "All the other ruminants occupy a relatively similar morphospace" The morphospace is multivariate and the authors are here referring to bgPC2 only. Could they rephrase possibly referring to the fact that these ruminant groups overlap along bgPC2 scores?

-L316: "the posterior semicircular and the anterior semicircular" There is no need to repeat here neither "semicircular" nor "the" in my opinion.

-L339: I do not understand why the authors are referring to "the total information". The CVs capture the "variance", as PCs and bgPCs do in a PCA and bgPCA, respectively.

-L348-351: Here the authors might have mistakenly reported again the results for bgPC2 (it seems like a copy paste gone wrong as it has the same wording as above). Could the please amend and check if they need to add information about the distribution of the taxa along the CV2 (not bgPC2)? Also, please take into account my comment about the wording mentioned in the same part for bgPC2 at L313.

-L352: the authors should remove the "a" before "longer"

-L357: Since the authors are implicitly (in my opinion) referring to the taxa included inside the Antilocapridae family, I think that they should not use the verb in the spelling for the 3rd singular person.

-L374: did the authors mean "long-lasting" instead of only "lasting" here?

-L779: the authors seem to have inadvertently skipped a point I raised regarding the Table S1, which was as follows. Could the authors provide pixel resolution for each of the specimens listed in Table S1?

Figures S5: in the response to a previous comment of mine, the authors state that they have fixed the visualization for the numbers near the fixed landmarks for this figure. I do not know if they inadvertently uploaded again the previous version of the figure, but the numbers are still not showing in the version I was able to download from the reviewing panel.

Figures in Supplementary Data 1 – RPANDA folder: in all instances the figures in the PDF files displays "Times" as the title for the X axis, while it should read as "Time".

Figures in Supplementary Data 1 –bgPCA_191 folder: the extreme morphologies (both PDF and .png) reported for this analysis appear to be inadvertently repeated (i.e., Antilo-dl is the same as Antilo-do; the same applies to several other cases in the folder). Could the authors check and fix this issue? Also, why are not they named following the extreme (maxbgPC1/minbgPC1, maxbgPC2/minbgPC2) they represent as in the case of the PCA?

Reviewer #2:

Remarks to the Author:

The manuscript changed distinctively during the revision, and I personally like the direction it was taken towards. Yet, to me, several issues remain that I would want to see resolved before publication. I guess it would be fair to the authors if someone else judges whether my comments are to the point or not.

1. The whole setup of the material.

1.1 There is an enormous amount of supplementary data that is, at least for someone of my intelligence and patience, badly organized. One does find things in the end, but they are not sorted intuitively, the setup (e.g., colour-coding, e.g. in the different rpanda graphs) is not consistent, the references in the main text are just to the data supplement folder that contains many sub-folders and I have to search for the information in there; sometimes there are several results given in there (for 306 specimens, or for 191 species) and one does not know which one the text refers to ... If I want to take this work seriously, it takes a lot of effort to find my way through the material.

1.2 If both results are given 191/306, then I would expect at least some supplementary text explaining this and what the difference means.

1.3 I believe that some of the material provided is not useful, e.g. the rpanda plot based on bg-PC, so to some extent, it feels as if just every possible combination of analyses was provided but not with text that explains the choices or that interprets the corresponding result. I would like to understand why the rpanda plot based on CVA seems to yield results similar to the PCA-based one, in contrast to the bg-PCA-based one.

1.4 The folder bg_PCA_306 seems empty, some results are missing there. Hence, I could only look at the bg_PCA_191 plot, which is in my understanding not in accord with the description of the by_PCA in the results text.

1.5 The gifs make the 3d-plots more accessible, but I had the feeling they are not identical to the graphs in the main text but 'mirrored'. That is disturbing at first until one learns to accept it - if that cannot be mended, this should be stated somewhere. Also, it takes a while to understand why you have 2 folders of gifs.

2. Because the tone of the manuscript has changed, and BL shape is no longer promoted as superior to some other measure (my apologies if I got the wrong impression first time round), the whole text flow is ok in my view now. But the main issue I had before remains: what is the difference to doing the same analyses just using body mass or some other proxy? And together with another reviewer we wondered about the effect of having different numbers of species per clade, i.e. whether the species number alone would also affect the results. Maybe the latter question could just be dealt with by mentioning it (rather than doing sensitivity analyses by cropping the number of species), the former question still holds.

2.1 The authors write very briefly:

"(BL centroid size) has been used here to explore size evolution and diversification in ruminants. The results of these analyses are given in Supplementary Data 1. We find that they differ significantly from results based on BL morphological evolution (evolutionary rate and significant shifts, as well as large-scale correlation with environmental factors) ..."

For me, these differences are important. First, this result is not easy to find in the supplementary data (see comment 1). Then, as mentioned, the evolutionary rate is of a very different magnitude (and for the CVA-based rpanda approach, we have again a different magnitude yet some similarity of shape).

2.2 But as for the large-scale correlation with environmental temperature, I disagree, for the the correlations seem to look really similar. A table that gives the correlations for each clade for the BL shape and the BL centroid size with temperature at one glance would be good to convince folks like me, or to check whether there is really a large difference. I predict, from the graphs, that there is not.

2.3 So what does this mean? Why should we believe the one based on BL shape more than the one based on BL centroid size? But in my view, if I look at the patterns of evol. rate change, it seems to me that within groups, the patterns are rather quite similar in between the two approaches. In my view, this would have to be clearly explained and discussed. The fact that the groups rank differently when using a size proxy as compared to the size-free shape proxy - should that not warn against using the shape proxy?

2.4 As the main part of the discussion appears to me to explain a scenario I thought well-known (but I admit I cannot provide a reference quickly, I am not a paleontologist), realizing the difference between a shape-based approach to evolutionary patterns and a size-based approach is important. If both tell the same story, that would make the story stronger. If they differ, then as a reader I would

be interested in the differences.

3. just as a side note, given that there will be many more (fossil) species in some groups, e.g. the tragulids or the giraffids, how valid are generalized statements about these groups? Maybe a modifier phrase would be good in this respect.

I made some other comments in the attached manuscript.

sincerely marcus clauss

Reviewer #3:

Remarks to the Author:

The authors have addressed all reviewer's comments well, and the manuscript is now clearer and more "punchy" than it was before. The analyses have a better rationale and the figures are clearer. There are still some minor grammatical issues and bumpy areas (e.g. "The PCA polymorphospace seems to represent a phylomorphospace" could be changed to something clearer), but these should be picked up by the copy editors.

The one area I would suggest more improvement is the discussion. As with the re-write of the Intro, this is a stylistic issue and in no way structural. The discussion is very results-focused (the first sentence is a results sentence with statistics, which do not belong in a discussion) and does not reference the introduction enough. An overall statement on the results and how these move the field forward could go to the beginning of the manuscript (e.g. "We set out to ask [summary of questions]. Our results show [summary of highlights]. Each paragraph could start and end with more context (e.g. Our expectation that X was confirmed by [brief, nonstatistical results summary]). I look forward to the paper coming out and the media flurry around it.

It is good to hear that the 3D files will also ultimately be made public.

Vera Weisbecker

REVIEWER COMMENTS

Reviewer #1 (Remarks to the Author):

The revised version of the manuscript by Mennecart and coauthors titled "Hidden in the deep: ruminant inner ear sensory organs record 35 million years of major historical events" shows a great improvement in its clarity and in the way the results are presented to the reader. Particularly, the structure is now much easier to be followed and guides much better the reader through the very interesting conclusions reached by the authors. I am more than satisfied by the way the authors dealt with the suggestions provided and implemented them into the new version of main text. I think that the manuscript will be perfectly suitable for publication as soon as the authors fix a few quite minor points (see below).

Overall, I consider that this manuscript will be a key contribution in inner ear studies, especially as it makes clear that some of the most-commonly accepted paradigms (e.g., the relationship between canal shape and locomotion) should be considered more carefully than what previously suggested. A job well done, congratulations!

Alessandro Urciuoli

Thank you very much for your comments.

As a general minor comment regards the use of the Supplementary Data 1. The authors often refer to it throughout the text. However, the compressed folder is actually quite large and includes a lot of very valuable information. Would there be a different manner to cite this information that could more clearly state to which of the files the authors are referring to each time?

We have simplified the Supplementary Data 1 (all that was not cited in the text has been deleted) and reorganized it into subfolders:

Supplementary_material_1-1 Raw dataset and R code

Supplementary_material_1-2 Geometric morphometrics

Supplementary_material_1-3 RRphylo

Supplementary_material_1-4 RPANDA

We now cite in the MS which subfolder is concerned.

The rest of the suggestions can be found on the merged PDF with tracked changes with the following line references:

-L86-87: The sentence starting with "Body mass..." seems to be misplaced between two others that

deal with the issues related with the fragmentary nature of the fossil record. Possibly the authors could move it right after the sentence ending at L85 with "...evolutionary rates"?

done

-L119: I think that "the" is missing in between the words "to evolution"

done

-L194: even if I understand that "cochlea" is used here as an adjective, the wording results somewhat difficult to be followed. As an alternative, the authors could use "first cochlear turn thickness".

done

-L216: I would suggest the authors to change "in" for "of"

done

-L262: when answering to a point I raised for the previous version of the manuscript, the authors are stating that "the PCA polymorphospace seems to represent a phylomorphospace". What do the authors mean? A phylomorphospace is just obtained by projecting a phylogenetic tree onto a morphospace, regardless of the observed distribution of the tips in the morphospace. To my understanding, the permutation test ($p < 0.001$) that they have performed suggest that the average amount of shape change along the branches of the tree is relatively small due to the presence of phylogenetic signal (based on the definition of the test by Klingenberg and Gidaszewski).

we have changed for:

The PCA polymorphospace **possesses a strong phylogenetic** signal (permutation test $p < 0.001$). **The average amount of shape change along the branches of the tree is relatively small due to the presence of a phylogenetic signal.**

-L266: As per its design, the PCA accumulates the greatest amount of variance on the first PC. I would suggest the authors to directly refer to the amount of variance explained by this PC.

done

-L276: Missing word capitalization within the parentheses.

done

-L281: I would suggest the authors to change "anterior" for "anteriorly" and to add the article "the" right before "posterior"

done

-L291: the authors here refer to the fact that Bovidae is the most variable group among those considered. Could the authors provide a very brief recapitulation sentence stating which are the extremes of this range of variation (e.g., "including species with both very short and very long cochlea")? If the authors consider that this might be redundant, please ignore this comment.

This sentence has been deleted.

-L313: "All the other ruminants occupy a relatively similar morphospace" The morphospace is

multivariate and the authors are here referring to bgPC2 only. Could they rephrase possibly referring to the fact that these ruminant groups overlap along bgPC2 scores?

done

-L316: “the posterior semicircular and the anterior semicircular” There is no need to repeat here neither “semicircular” nor “the” in my opinion.

done

-L339: I do not understand why the authors are referring to “the total information”. The CVs capture the “variance”, as PCs and bgPCs do in a PCA and bgPCA, respectively.

Total information has been changed for variance

-L348-351: Here the authors might have mistakenly reported again the results for bgPC2 (it seems like a copy paste gone wrong as it has the same wording as above). Could the please amend and check if they need to add information about the distribution of the taxa along the CV2 (not bgPC2)? Also, please take int-o account my comment about the wording mentioned in the same part for bgPC2 at L313.

Indeed sorry it has been a bad copy past erasing the right version of the text. It can also see from the break in the text. It has been changed into:

Along CV2 axis (16.9%), the negative values are mostly occupied by Bovidae, while Stem Ruminantia occupy the highest values (Supplementary Data 1-2). A trend can be observed with the Giraffomorpha and the Cervidae having the lower CV2 positive scores, then the Moschidae and Antilocapridae having slightly higher CV2 scores, and finally the Stem Pecora having just lower CV2 scores than the Stem Ruminantia.

-L352: the authors should remove the “a” before “longer”

done

-L357: Since the authors are implicitly (in my opinion) referring to the taxa included inside the Antilocapridae family, I think that they should not use the verb in the spelling for the 3rd singular person.

done

-L374: did the authors mean “long-lasting” instead of only “lasting” here?

done

-L779: the authors seem to have inadvertently skipped a point I raised regarding the Table S1, which was as follows. Could the authors provide pixel resolution for each of the specimens listed in Table S1?

This has been done.

Figures S5: in the response to a previous comment of mine, the authors state that they have fixed the visualization for the numbers near the fixed landmarks for this figure. I do not know if they inadvertently uploaded again the previous version of the figure, but the numbers are still not showing in the version I was able to download from the reviewing panel.

Sorry I have uploaded the wrong version of the figure. It has been corrected now.

Figures in Supplementary Data 1 – RPANDA folder: in all instances the figures in the PDF files displays "Times" as the title for the X axis, while it should read as "Time".

It has been changed

Figures in Supplementary Data 1 –bgPCA_191 folder: the extreme morphologies (both PDF and .png) reported for this analysis appear to be inadvertently repeated (i.e., Antilo-dl is the same as Antilo-do; the same applies to several other cases in the folder). Could the authors check and fix this issue? Also, why are not they named following the extreme (maxbgPC1/minbgPC1, maxbgPC2/minbgPC2) they represent as in the case of the PCA?

Indeed, there was a problem in the routine that has been changed. The folders for the bg-PCA and the CVA contains mean shapes and not extreme shapes. The folders are now renamed correctly.

Reviewer #2 (Remarks to the Author):

The manuscript changed distinctively during the revision, and I personally like the direction it was taken towards. Yet, to me, several issues remain that I would want to see resolved before publication. I guess it would be fair to the authors if someone else judges whether my comments are to the point or not.

1. The whole setup of the material.

1.1 There is an enormous amount of supplementary data that is, at least for someone of my intelligence and patience, badly organized. One does find things in the end, but they are not sorted intuitively, the setup (e.g., colour-coding, e.g. in the different rpanda graphs) is not consistent, the references in the main text are just to the data supplement folder that contains many sub-folders and I have to search for the information in there; sometimes there are several results given in there (for 306 specimens, or for 191 species) and one does not know which one the text refers to ... If I want to take this work seriously, it takes a lot of effort to find my way through the material.

We have reorganized the Supplementary Data 1 into subfolders:

Supplementary_material_1-1 Raw dataset and R code

Supplementary_material_1-2 Geometric morphometrics

Supplementary_material_1-3 RRphylo

Supplementary_material_1-4 RPANDA

We now cite in the MS which subfolder is concerned.

We have deleted the supplementary that are not used in the text (being similar to the original data when using RRphylo and RPANDA on PCA, CVA, and bg-PCA).

1.2 If both results are given 191/306, then I would expect at least some supplementary text explaining this and what the difference means.

This is already explained in the text in the first line of the Material and Method part:

“We selected and scanned petrosal bones, which house the BL, from 306 specimens representing 191 ruminant species from the early Oligocene (ca. 33 Ma) to the present (Table S1).”

We have added in the phylogenetic part that 191 correspond to the number of species

A phylogenetic tree was obtained using Mesquite 3.04 software³⁹ combining specific phylogenetic hypothesis of the 191 species into a combined tree

We have added in the *Shape variation and phylogenetic signal* part that 191 correspond to the number of species and 306 are all the specimens including intraspecific variability

When several specimens of a species were available, we created a mean shape for the species using the function “*mshape*” of the package *geomorph* v.4.0.3^{36,37} for the analyses that are using **only one specimen per species** in a phylogenetic framework (**dataset with 191 species**). We performed a Principal Component Analysis (PCA) computed using the function “*procSym*” of the R package *Morpho* v2.9⁶⁵ to study the shape variation within the dataset in its natural scale, **including intraspecific variation (dataset with 306 specimens)**.

1.3 I believe that some of the material provided is not useful, e.g. the rpanda plot based on bg-PC, so to some extent, it feels as if just every possible combination of analyses was provided but not with text that explains the choices or that interprets the corresponding result. I would like to understand why the rpanda plot based on CVA seems to yield results similar to the PCA-based one, in contrast to the bg-PCA-based one.

We understand why the reviewer made this comment, and we have deleted the graphs that were not used in the text. There were some similarities because in both cases this was an analysis based on the shape (the basic data are the same). One is showing the evolution of shape variation through time (PCA), while the CVA most likely shows more about cladogenesis since we are maximizing shape similarities within clades. Nevertheless, even if some similarities can be seen in the general trends, large differences are also present in the amplitude and trends (e.g. Bovidae and Moschidae).

1.4 The folder bg_PCA_306 seems empty, some results are missing there. Hence, I could only look at the bg_PCA_191 plot, which is in my understanding not in accord with the description of the by_PCA in the results text.

We have deleted bg_PCA 191 that we are not talking about in the text. Sorry for the inconvenience.

1.5 The gifs make the 3d-plots more accessible, but I had the feeling they are not identical to the graphs in the main text but 'mirrored'. That is disturbing at first until one learns to accept it - if that

cannot be mended, this should be stated somewhere. Also, it takes a while to understand why you have 2 folders of gifs.

We are not certain if we understand correctly, here. This is an animated .gif. Indeed, if you open it in a raster graphics editor, you will have only one view (Paint, Photoshop, word). You should try with Preview, or opening with edge, explorer, or in the viewer of various software or website.

2. Because the tone of the manuscript has changed, and BL shape is no longer promoted as superior to some other measure (my apologies if I got the wrong impression first time round), the whole text flow is ok in my view now. But the main issue I had before remains: what is the difference to doing the same analyses just using body mass or some other proxy? And together with another reviewer we wondered about the effect of having different numbers of species per clade, i.e. whether the species number alone would also affect the results. Maybe the latter question could just be dealt with by mentioning it (rather than doing sensitivity analyses by cropping the number of species), the former question still holds.

We have families with only one species (antilocapridae) where we can observe an increase in the rates but other long branches with only one species (e.g., Hyemoschus), where a decrease is observed. Within duikers, we can observe a clade with an acceleration and another one with a deceleration. These are rates, so the number of species does not influence the result of the RPANDA analysis. In RRphylo this is using the direct data, mapping on the tree.

2.1 The authors write very briefly:

"(BL centroid size) has been used here to explore size evolution and diversification in ruminants. The results of these analyses are given in Supplementary Data 1. We find that they differ significantly from results based on BL morphological evolution (evolutionary rate and significant shifts, as well as large-scale correlation with environmental factors) ..."

For me, these differences are important. First, this result is not easy to find in the supplementary data (see comment 1). Then, as mentioned, the evolutionary rate is of a very different magnitude (and for the CVA-based rpanda approach, we have again a different magnitude yet some similarity of shape).

We have cleaned the sup data. We have developed this part (see 2.3) and directly add to the supplementary figures a figure with the RRphylo of the PCA and centroid size evo rates to show graphically that the statistically significant shifts are not the same (Fig. S4).

Figure S4. Comparative evolutionary rates of the ruminant BL morphology based on PC scores (a) and of the size based on the centroid size (b) through the phylogenetic tree. Significant decrease in the evolutionary rates are marked by a blue circle, while a significant increase of the evolutionary rates is marked by a red circle. We can observe that number of significant shifts, their location in the tree, and the nature of the shift (significant decrease or increase of the evolutionary rate) differ when considering the evolutionary rates of the ruminant BL morphology and of the size based on the centroid size. Methodology and statistical results produced by the R packages RRphylo⁷³ and Phytools⁷⁰ are provided in Material and Methods and Supplementary Data 1-3. Silhouettes of the families modified from⁹⁶. Same color code as in Figure 2 for ages.

2.2 But as for the large-scale correlation with environmental temperature, I disagree, for the the correlations seem to look really similar. A table that gives the correlations for each clade for the BL shape and the BL centroid size with temperature at one glance would be good to convince folks like me, or to check whether there is really a large difference. I predict, from the graphs, that there is not.

The curves of the different families are clearly different. On the centroid size, giraffes are increasing at the very end by a huge magnitude, and all the others (except moschids) are doing the same but at a smaller magnitude.

The statistical results are provided in Supplementary_material_1-4 RPANDA.

Here are the results from the PCA (Supplementary_material_1-4 RPANDA/PCA)

	Tragulina	Antilocap.	bovidae	cervidae	moschidae	giraffidae
aic	-94.027238	-54.378731	-456.45308	-263.77496	-67.096927	-55.718262
aicc	-91.360571	-48.378731	-456.16393	-263.20353	-62.296927	-52.718262
likelihood	50.0136189	30.1893654	231.226542	134.88748	36.5484635	30.8591309

Here are the results from the centroid size (Supplementary_material_1-4 RPANDA/centroid)

	Tragulidae	Antilocap.	bovidae	cervidae	moschidae	giraffidae
aic	23.3549871	16.5059894	300.024841	144.70344	42.9740441	55.1620376
aicc	35.3549871	22.5059894	300.313998	145.274868	47.7740441	58.1620376
likelihood	-8.6774935	-5.2529947	-147.01242	-69.35172	-18.487022	-24.581019

The statistical results are different.

2.3 So what does this mean? Why should we believe the one based on BL shape more than the one based on BL centroid size? But in my view, if I look at the patterns of evol. rate change, it seems to me that within groups, the patterns are rather quite similar in between the two approaches. In my view, this would have to be clearly explained and discussed. The fact that the groups rank differently when using a size proxy as compared to the size-free shape proxy - should that not warn against using the shape proxy?

We present statistical results of different analyses. They do explain different scenarios since the original datasets (shape vs size) are different. BL evolves more neutrally than size. Size is highly correlated to ecology and we aimed at understanding what factors besides ecological parameters could drive the evolution ruminants. The results we got for the size seems to indicate a Depéret-Cope's rule with an increase of the evolutionary rates of the size in younger species. We added more explanation in the text.

To compare the impact of the topology of the tree on the evolutionary results, similar analyses have also been performed using the size of the specimens based on the BL centroid size. BL size has been proven to be a good proxy for body mass estimation in ruminants⁷⁶ and has been used here to explore size evolution and diversification in ruminants. Contrary to the BL morphology, that may reflect a neutral evolution³⁰, size is known to be strongly correlated with ecological factors like diet and local environment. Size thus provides other information than BL morphology. The results of these analyses are given in Fig. S4, Supplementary Data 1-3, and Supplementary Data 1-4. We find that evolution of size differs significantly from results based on BL morphological evolution (evolutionary rate and significant shifts in Fig. S4, as well as large-scale correlation with environmental factors; Supplementary Data 1-3 and 1-4), which indicates that the topology of the tree and the number of

considered taxa are not the main factor driving our results. An increase in the size evolutionary rate is observed through time in most of the clades (Supplementary Data 1-4), a scenario that may be expected considering the Depéret-Cope's rule. Evolutionary rates of the BL are more heterogeneous and explained in detail in the following.

2.4 As the main part of the discussion appears to me to explain a scenario I thought well-known (but I admit I cannot provide a reference quickly, I am not a paleontologist), realizing the difference between a shape-based approach to evolutionary patterns and a size-based approach is important. If both tell the same story, that would make the story stronger. If they differ, then as a reader I would be interested in the differences.

Indeed, they differ because we are not studying the same thing. With a BL evolving neutrally, we study the broad evolution of the clade, the impact of intrinsic and extrinsic factors on this evolution. When studying size, we are focusing on one ecological parameter that is highly correlated to the environment and also influenced by the Depéret-Cope's rule. Moreover, size may more suffer from sampling effect than morphological evolution because the BL has a gradual change. It is really good for us that the results are different using different datasets, and explaining different parts of the broad story. But the purpose of this article is to focus on the broad evolution of the ruminants and to understand it using a unique object that has never been used for that, the BL. Using only 16% of the known diversity, we were able to find similar results to what we may obtain using the 2000 species multi parameters at once. This shows how powerful BL morphology is as a tool for reconstructing mammalian evolution.

3. just as a side note, given that there will be many more (fossil) species in some groups, e.g. the tragulids or the giraffids, how valid are generalized statements about these groups? Maybe a modifier phrase would be good in this respect.

We have five tragulid species out of 30 known (16.5%) and in total 190 species of ruminants in our analysis out of the 1200 species known today (16%; both extant and extinct), so, relatively, the Tragulidae are sampled like the rest. For the Bovidae, for which we have 100 species, this represents ca. 20% of their known diversity, which is about the same as the Tragulidae. We agree that the tree is undersampled considering the broad evolution of the ruminants (furthermore, new species are being constantly described), but we got extremely interesting results in this respect!! We were able to reconstruct the broad view of the ruminant evolutionary history thanks to the evolution rate of the BL with only 16% of the known diversity. We can see where and when there is an acceleration/deceleration of the evolution. For sure, including *Dorcabune* in the Tragulidae phylogeny may give more information, or even provide a different story for this specific lineage within tragulids, but this clade is supposed to be more basal than *Dorcatherium*, so its evolutionary rate would not affect the tragulid clade we are considering here and the story we are deciphering.

We have added in the text:

The resulting dataset encompasses the most extensive study to date using original data of the inner ear region representing 16% of the all known ruminant diversity including 16.5% of the Tragulidae and ca. 22% of the Bovidae and of the Cervidae.

I made some other comments in the attached manuscript.

Your comments in the attached manuscript were taken into consideration. Thank you.

Discussion

all what is written here is correct -- but I thought that all this was more or less what you find when reading about ruminant evolution.

It would be good to explain why / if the picture would be different when using BL centroid size, or whether the same narrative would evolve.

It is fair to say, given this text here, that BL shape confirms what we think about ruminant evolution. I am not saying a "added value" of BL shape is compulsory. I just feel, when reading this, that I would like a statement like "little added value" or "added value" (the latter e.g. by comparing against BL centroid size scenario).

We have added this sentence to recall that we know already about the ruminant evolution. Indeed, we get similar results (fortunately), but here only with 16% of total ruminant diversity represented, using an innovative method that may well become a hallmark in evolutionary studies because inner ears are virtually available for thousands of fossil mammalian species and seem to be a very powerful tool indeed.

Rates of evolution show radiation events

Ruminants underwent several episodes of radiations in relation to environmental changes^{20,31,47}.

Morphological evolution correlates to climate and geological events

Morphological traits like hypsodonty and appendage characters have been shown to be correlated to environmental changes³¹.

We would like to mention that we really appreciate your review. It really helped us to structure the MS and we really hope that we succeed to make it accessible, clear, and convinced you of the interest of this study.

sincerely marcus clauss

Reviewer #3 (Remarks to the Author):

The authors have addressed all reviewer's comments well, and the manuscript is now clearer and more "punchy" than it was before. The analyses have a better rationale and the figures are clearer. There are still some minor grammatical issues and bumpy areas (e.g. "The PCA polymorphospace seems to represent a phylomorphospace" could be changed to something clearer), but these should be picked up by the copy editors.

Thank you very much. We changed it for:

"The PCA polymorphospace possesses a strong phylogenetic signal (permutation test $p < 0.001$). The average amount of shape change along the branches of the tree is relatively small due to the presence of a phylogenetic signal."

The one area I would suggest more improvement is the discussion. As with the re-write of the Intro, this is a stylistic issue and in no way structural.

We made introductions to the discussion echoing the introduction.

The discussion is very results-focused (the first sentence is a results sentence with statistics, which do not belong in a discussion) and does not reference the introduction enough.

An overall statement on the results and how these move the field forward could go to the beginning of the manuscript (e.g. "We set out to ask [summary of questions]. Our results show [summary of highlights]).

We have ended the introduction by this additional paragraph

In this study, we set out to ask if the BL is a good proxy to study the phylogeny and evolution of a whole mammalian clade, directly including fossil data into the analysis. Since BL morphology may partly evolve neutrally and since size evolution is more related to ecological factors, a comparison between both is likely to illuminate different aspects of ruminant evolution. In addition, we here question if BL morphological evolutionary rates help understand the relationships between intrinsic and extrinsic factors shaping the evolutionary processes. Our results show that BL shape morphology compares well to molecular-based phylogenetic hypotheses, with great similarities in the results. BL morphology and size do display different evolutionary scenarios. While size seems to follow the Depéret-Cope's rule, with an increase of size evolutionary rates in younger taxa, BL morphological evolutionary rates through time change differently depending on the considered clade. BL morphological evolutionary rates clearly respond to intrinsic factors like ecological opportunity combined with environmental and global to regional geographical conditions.

Each paragraph could start and end with more context (e.g. Our expectation that X was confirmed by [brief, nonstatistical results summary]).

Inner ear shape reflects phylogeny

As expected according to previous analyses of the BL in vertebrates^{20,25,77-79}, the impact of the phylogeny on the shape of the BL is significant in ruminants.

Rates of evolution show radiation events

Ruminants underwent several episodes of radiations in relation to environments changes^{20,31,47}. The evolutionary rates of the BL morphology reveal different scenarios for the different clades of ruminants.

Morphological evolution correlates to climate and geological events

Morphological traits like hypsodonty or appendage characters have been shown to be correlated to environmental changes³¹. This is not systematically the case considering the morphological evolution of the ruminant BL.

I look forward to the paper coming out and the media flurry around it.

It is good to hear that the 3D files will also ultimately be made public.

Vera Weisbecker

Thank you very much for your helpful comments

Reviewers' Comments:

Reviewer #1:

Remarks to the Author:

The last revised version of the manuscript NCOMMS-21-44912B and titled "Hidden in the deep: ruminant inner ear sensory organs record 35 million years of major historical events" has further increased its readability and clearness. Particularly, I found most useful the implementation of a different structure for the supplementary data, which has greatly eased the references to this useful data in the main text. Following these improvements and the already good quality of the previous revised version of the manuscript, I have just a few very minor considerations to provide to the authors. These are the following (line numbers refer to the track changes version of the manuscript):

L269-270 - "size is known to be strongly correlated with ecological factors like diet and local environment" Could the authors provide a reference for this statement?

L277 - "Depéret-Cope's rule" could the authors provide a short sentence explaining the effects fo the Depéret-Cope's rule? Some readers might not be familiar with it.

L283-285 - "The average amount of shape change along the branches of the tree is relatively small due to the presence of a phylogenetic signal" I might not be understanding what the authors are referring to here. Do they mean that the amount of shape change along the phylogeny branches is inversely proportional to the amount of phylogenetic signal? Could the provide a short explanation about why is this the expected outcome for evolutionary rates when there is a phylogenetic signal?

L327 - "...clades observed..." I think it should read "...clades is observed..."

Alessandro Urciuoli

Reviewer #2:

Remarks to the Author:

I thank the authors for implementing so many changes due to the reviewers' comments.

I know I cannot contribute further than the following comments and ask that I be replaced from here on as reviewer. I list my concerns here and have attached the annotated manuscript and reply letter.

1. I still think the supplementary material is not organised well. For example, sometimes it is called «supplementary material», sometimes it is called «supplementary data» - again, it took me a while to understand the different words mean (I believe) the same thing. Labelling of the individual files in these folders is not consistent (e.g. there is one file starting with «AIC» in the «Rpanda_PCA»). Several of the excel files have the data output from R copied into them without this output being converted to columns, and the column headings are not consistent.

2. I realize that I do not understand how Akaike is used. I do not think the wording is correct, but not sure. Akaike does NOT test significance but model fit! I tried to understand the corresponding tables but failed.

In my view, you cannot test significance using Akaike.

Akaike works to compare different models, in my view (so you have to have more than one). But what are the different models you are using here? Maybe you use a variety of models for each family - like smoothing level 1 vs. smoothing level 3 vs. smoothing level 4? This would mean you can check, for each species, which smoothing level gives the best fit. But what does that tell you?

But why would you do that for each family separately?

But then, you provide a table like the ones used in the reply letter, and they do not make sense ... they give, for each family, only a single Akaike value. That makes no sense, as then you do not compare the Akaike of the model against any other model? Why do Akaike for a single model?

In these tables, you have different Akaike for each family - this means you run individual models for each family (because you cannot have several Akaike values within a single model).

To sum up, I do not understand what was done here.

3. Neutral evolution:

Probably ignorance on my side: somehow, saying in the abstract there is neutral evolution (actually, you state in the abstract that you SHOW that there is neutral evolution) and then saying there is a relationship to climate seems contradictory to me. If there is a clear response to some external factor proxy, is that really neutral evolution?

I suspect that you ASSUME neutral evolution for the inner ear (which makes sense as we cannot think of a logical way to link it to climate) and then use it as a proxy for speciation. Not sure if I am right but just re-consider the wording here and just overrule me if I am wrong. If you say you "show" BL follows neutral evolution, you have to outline a concept of how to prove neutral evolution.

4. Body size analyses

This is my main area of confusion / concern: I urge the authors to re-consider what they write and show here, because in my view, this might backfire on them.

(a) the link to the Déperet-Cope's rule. I do not understand the sentence, my apologies. The evolutionary RATE of size change - in my understanding, this does not have a direction (e.g., towards larger size) - the evolutionary rate just indicates change itself. So, a high rate is not directly linked to a shift towards large body size but to body size CHANGE in my view. The rate is high = there is change (in any direction).

I do not believe that the DC rule can have anything to do with a change in evolutionary rate ... because the rule does not say, changes become more intense in more recent times. Or does it? I thought the rule says something like "there is a constant increase in size" (which would NOT translate into ANY change in the evol. rate in my view - on the contrary, evolutionary rate of body size would just be a constant).

(b) the difference to the BL evolutionary rates. What you are saying is that when ruminants change in size (evol. change), this happens without a change in BL! And when they change in BL, they do not change in size. Do you really believe that? This would mean BL is not 'neutral' but actually conserved across body sizes. There are many parts in the narrative in the discussion that completely ignore this seeming discrepancy. If body size evolution occurs via speciation, and BL follows a neutral evolution = just reflects speciation, I would expect the two evolutionary rates to be really similar. Any deviation would have to be explained.

This leads me to a visual suspicion:

(c) Is the data displayed correctly? I have one question here and apologize again if it is stupid. If you write there is an increase in younger taxa in all clades in the evolutionary rates of size ... with the highest change in the giraffes? Is that correct?

are you sure the graphs you put into the rebuttal, and that are found as

RPANDA_pca_temp_Trag-Fam

and

RPANDA_cent_temp_Trag-Fam

do not have a confusion in how the taxa are labelled? I am not saying there is a mistake, I just ask you to check it once more -- for the centroid size, the Moschidae yellow line looks so identical to the line for the giraffids in the PCA graph. And light blue line of the giraffe in the centroid size plots looks to similar to the bovidae or cervidae line in the PCA graph. This may be coincidence, but just make sure. See also my comments in the reply letter on this.

sincerely marcus clauss

Reviewer #3:

Remarks to the Author:

Thank you for adding the extra contextualisation to the discussion, it now reads much better. I am happy with the manuscript as it is now.

Reviewer #1 (Remarks to the Author):

The last revised version of the manuscript NCOMMS-21-44912B and titled "Hidden in the deep: ruminant inner ear sensory organs record 35 million years of major historical events" has further increased its readability and clearness. Particularly, I found most useful the implementation of a different structure for the supplementary data, which has greatly eased the references to this useful data in the main text. Following these improvements and the already good quality of the previous revised version of the manuscript, I have just a few very minor considerations to provide to the authors.

Thank you very much !!

These are the following (line numbers refer to the track changes version of the manuscript):

L269-270 - "size is known to be strongly correlated with ecological factors like diet and local environment" Could the authors provide a reference for this statement?

We have added these 3 references

Clauss M., Steuer P., Müller D.W.H., Codron D., & Hummel J. Herbivory and Body Size: Allometries of Diet Quality and Gastrointestinal Physiology, and Implications for Herbivore Ecology and Dinosaur Gigantism. *PLoS ONE* **8(10)**, e68714 (2013)

du Toit J. T & Owen-Smith N. Body size, population metabolism, and habitat specialization among large African herbivores. *Am. Nat.* **133(5)**, 736–740 (1989)

Mennecart B., Becker D., & Berger J.-P. Mandible shape of ruminants: between phylogeny and feeding habits. In: *Ruminants: Anatomy, behavior, and diseases*, Mendes R. E., Nova Science Publishers. pp. 205–226 (2012).

L277 - "Depéret-Cope's rule" could the authors provide a short sentence explaining the effects fo the Depéret-Cope's rule? Some readers might not be familiar with it.

We have added:

An increase of the maximum size, and of the evolutionary rate of the size, is observed through time in most of the clades (Fig. S5). The tendency of evolutionary lineages to increase their body size through time is known as the Depéret-Cope's rule⁸². An increase in the evolutionary rates of the size is already known in insular context and during radiations⁸³.

L283-285 - "The average amount of shape change along the branches of the tree is relatively small due to the presence of a phylogenetic signal" I might not be understanding what the authors are referring to here. Do they mean that the amount of shape change along the phylogeny branches is inversely proportional to the amount of phylogenetic signal? Could the provide a short explanation about why is this the expected outcome for evolutionary rates when there is a phylogenetic signal?

We used your sentence of the revision 2

-L262: when answering to a point I raised for the previous version of the manuscript, the authors are stating that "the PCA polymorphospace seems to represent a phylomorphospace". What do the authors mean? A phylomorphospace is just obtained by projecting a phylogenetic tree onto a morphospace, regardless of the observed distribution of the tips in the morphospace. To my understanding, the permutation test ($p < 0.001$) that they have performed suggest that the average amount of shape change along the branches of the tree is relatively small due to the presence of phylogenetic signal (based on the definition of the test by Klingenberg and Gidaszewski).
we have changed for:

The PCA polymorphospace possesses a strong phylogenetic signal (permutation test $p < 0.001$). The average amount of shape change along the branches of the tree is relatively small due to the presence of a phylogenetic signal.

We changed it for

The average amount of shape change along the branches of the phylogenetic tree is indeed very small, in comparison to the null hypothesis of no phylogenetic signal.

L327 - "...clades observed..." I think it should read "...clades is observed..."

We changed it for

Bg-PC1 axis represents 70.92% of the variance (Supplementary Material 1-2). A clear distinction between the Tragulidae and all the remaining ruminant groups can be observed.

Alessandro Urciuoli

Reviewer #2 (Remarks to the Author):

I thank the authors for implementing so many changes due to the reviewers' comments.

I know I cannot contribute further than the following comments and ask that I be replaced from here on as reviewer. I list my concerns here and have attached the annotated manuscript and reply letter.

1. I still think the supplementary material is not organised well. For example, sometimes it is called «supplementary material», sometimes it is called «supplementary data» - again, it took me a while to understand the different words mean (I believe) the same thing.

We changed Supplementary Data for Supplementary Material throughout the MS and renamed files.

Labelling of the individual files in these folders is not consistent (e.g. there is one file starting with «AIC» in the «Rpanda_PCA»). Several of the excel files have the data output from R copied into them without this output being converted to columns, and the column headings are not consistent.

The .csv files are easier to handle, smaller in size and read by many applications (not only excel). It is considered a standard format for data export and allows the user a great flexibility. Also, these files are directly usable in R. The .csv files are in fact created in R using an export function. It does transfer

the data into columns using separators (such as commas). We clarified the columns names of the documents for the RPANDA analyses but keep the .csv format for the export.

If you want to visualize the files in a more convenient way in Excel, select the first column, go to Data (Daten), Text in Spalten, and select the comma. The data are now converted in columns.

Please read the comment of reviewer 1:

“Particularly, I found most useful the implementation of a different structure for the supplementary data, which has greatly eased the references to this useful data in the main text.”

2. I realize that I do not understand how Akaike is used. I do not think the wording is correct, but not sure. Akaike does NOT test significance but model fit! I tried to understand the corresponding tables but failed.

In my view, you cannot test significance using Akaike.

Akaike works to compare different models, in my view (so you have to have more than one). But what are the different models you are using here? Maybe you use a variety of models for each family - like smoothing level 1 vs. smoothing level 3 vs. smoothing level 4? This would mean you can check, for each species, which smoothing level gives the best fit. But what does that tell you?

But why would you do that for each family separately?

Indeed we are comparing models, that is why we wrote Δ AIC in the text.

We changed the text for:

Each family having potentially a different ecological optimum and reacting differently to the environmental changes, we analysed separately each pecoran clade with the Tragulina. We then analysed the Pecora as a whole and the Tragulina in a second analysis. The significance of the results was tested against different levels of smoothing of the temperature curve, comparing to the resulting Aikake values. This implies that if the correlations are always significant in a clade, regardless of the smoothing parameter, the temperatures are not the only parameter influencing the evolutionary rates results. All supporting data concerning RPANDA are given in Supplementary Material 1-4.

We need to do it separately by families, because each family has a different ecological optimum and potentially react differently to the environmental changes. This is clearly observable, they have different evolutionary curves. In one case it very important to do it separately, the cervids. Smoothing is not that important because they always correlate with the temperature curves. It means not only the temperature has an impact, but another factor should be considered and this is the dispersal into south America: this is not the global cooling which made it possible to enter South America, but entering South America increased the evolutionary rate while the world was cooling down.

We also changed the name of the files in the supplementary material 1-4 to “AIC_results” instead of “significance” for clarification

But then, you provide a table like the ones used in the reply letter, and they do not make sense ... they give, for each family, only a single Akaike value. That makes no sense, as then you do not compare the Akaike of the model against any other model? Why do Akaike for a single model?

Indeed, you need to calculate it, the Δ AIC compares the different models. We provided all the information.

In these tables, you have different Akaikes for each family - this means you run individual models for each family (because you cannot have several Akaike values within a single model). To sum up, I do not understand what was done here.

We mention it in the methods, we applied different levels of smoothing of the temperature curve, to compare the correlation between the evolutionary rates and the temperature and if there is or not a significant correlation.

Each family having potentially a different ecological optimum and reacting differently to the environmental changes, we analysed separately each pecoran clade with the Tragulina and also analysed the Pecora as a whole with the Tragulina in a first analysis. We then analysed the Pecora as a whole and the Tragulina in a second analysis. The significance of the results was tested against different levels of smoothing of the temperature curve, comparing to the resulting Aikake values. This implies that if the correlations are always significant in a clade, regardless of the smoothing parameter, the temperatures are not the only parameter influencing the evolutionary rates results. All supporting data concerning *RPANDA* are given in Supplementary Material 1-4.

3. Neutral evolution:

Probably ignorance on my side: somehow, saying in the abstract there is neutral evolution (actually, you state in the abstract that you SHOW that there is neutral evolution) and then saying there is a relationship to climate seems contradictory to me. If there is a clear response to some external factor proxy, is that really neutral evolution?

We tried to make the sentence clearer :

This finding suggests that increasing seasonality related to decreasing temperature and coupled to changes in paleogeography promoted the development of heterogeneous habitats that fostered the Cervidae and the Bovidae diversification⁹⁻¹¹ in connection with their intrinsic ecological optimum.

This cannot be used as a proxy for climate. The evolution of the BL morphology is linked to the ecological optimum and the colonization of a new continent involving an ecological radiation, not directly to the climate. So the shape of the BL and its evo rates cannot predict past climate.

	Ecological optimum		
	Preference for cold Cervidae	Preference for tropical forest Tragulidae	Preference for intertropical environnements Giraffidae
cold period	Evo rate +	Evo rate =	Evo rate -
warm period	Evo rate -	Evo rate =	Evo rate +
colonisation of a new continent	Evo rate +	not observed	Evo rate +

I suspect that you ASSUME neutral evolution for the inner ear (which makes sense as we cannot think of a logical way to link it to climate) and then use it as a proxy for speciation. Not sure if I am

right but just re-consider the wording here and just overrule me if I am wrong. If you say you “show” BL follows neutral evolution, you have to outline a concept of how to prove neutral evolution.

The BL is a structure fully ossifying rather early in the development of mammal species. It is highly morphologically constrained. As mentioned in the text, neutral evolution of the BL is a not an entirely new concept, it's been hypothesized by others in recent times. We indeed assume it follows neutral evolution (and confirm it does) but we use it as a proxy for shape evolution not for speciation.

4. Body size analyses

This is my main area of confusion / concern: I urge the authors to re-consider what they write and show here, because in my view, this might backfire on them.

(a) the link to the Déperet-Cope's rule. I do not understand the sentence, my apologies. The evolutionary RATE of size change - in my understanding, this does not have a direction (e.g., towards larger size) - the evolutionary rate just indicates change itself. So, a high rate is not directly linked to a shift towards large body size but to body size CHANGE in my view. The rate is high = there is change (in any direction).

You are right, but we also have the centroid_size_phylo.pdf (/Supplementary_material_1-3 RRphylo/centroid) that shows that there is a general trend toward larger forms in younger species, hence the link to Depéret-Cope's rule. We would like to keep it as it is and added the figure in Figure S5.

I do not believe that the DC rule can have anything to do with a change in evolutionary rate ... because the rule does not say, changes become more intense in more recent times. Or does it? I thought the rule says something like “there is a constant increase in size” (which would NOT translate into ANY change in the evol. rate in my view – on the contrary, evolutionary rate of body size would just be a constant).

This is a logical way of putting it indeed, and what you write makes sense. But see here below where DC rule is detected in different clades with different evo rates of body size. This cited study (Bakker 2015) shows that accelerated evo rates significantly tend to produce increases in body size in many mammalian clades, following what the DC rule predicts:

<https://academic.oup.com/sysbio/article/65/1/98/2461670>

“Yet another approach is to assume that rates of evolution differ between branches of the phylogeny. Baker et al. (2015) made this assumption in their analysis of mammals, and did find evidence of Depéret's rule.”

<https://palaeo-electronica.org/content/2022/3663-body-mass-of-candiacervus>

“Remarkable is the speed under which this body mass range was achieved. Deer colonized Crete during the late Middle Pleistocene (van der Geer, 2018a). This means that within a relatively short time period, the Cretan deer evolved a remarkable body size diversity. Body size shifts, either towards dwarfing or gigantism, were shown to occur rapidly in other island mammals as well (e.g., Heaney, 1978; Lister, 1989, 1996; Millien, 2006; van der Geer, 2018b; van der Geer et al., 2018). It appears similarly rapid size changes occur, not just in anagenetic evolution, but in evolutionary

radiations as well.”

(b) the difference to the BL evolutionary rates. What you are saying is that when ruminants change in size (evol. change), this happens without a change in BL! And when they change in BL, they do not change in size. Do you really believe that? This would mean BL is not 'neutral' but actually conserved across body sizes. There are many parts in the narrative in the discussion that completely ignore this seeming discrepancy. If body size evolution occurs via speciation, and BL follows a neutral evolution = just reflects speciation, I would expect the two evolutionary rates to be really similar. Any deviation would have to be explained.

There may be a slight misunderstanding here: we work on shape. Correlation between shape and size is allometry. Indeed, we state here that there is little influence of the size on the shape of the BL (this is also why it is a good phylogenetic proxy). We have already published this idea elsewhere, stating that, excluding the open structures (endolymphatic sac, cochlear duct), the rest of the BL shows very few allometric growth (Billet et al. 2015 also came to the same conclusion).

BL morphology evolves neutrally and its evolutionary rates are linked to ecological optimum constraints.

BL size is linked to ecological parameters like feeding habit and habitat, it changes indeed between large and small species.

This leads me to a visual suspicion:

(c) Is the data displayed correctly? I have one question here and apologize again if it is stupid. If you write there is an increase in younger taxa in all clades in the evolutionary rates of size ... with the highest change in the giraffes? Is that correct?

are you sure the graphs you put into the rebuttal, and that are found as

RPANDA_pca_temp_Trage-Fam

and

RPANDA_cent_temp_Trage-Fam

do not have a confusion in how the taxa are labelled? I am not saying there is a mistake, I just ask you to check it once more -- for the centroid size, the Moschidae yellow line looks so identical to the line for the giraffids in the PCA graph. And light blue line of the giraffe in the centroid size plots looks to similar to the bovidae or cervidae line in the PCA graph. This may be coincidence, but just make sure. See also my comments in the reply letter on this.

We have reviewed several time our dataset and code. We checked your visual suspicion again this time and did not find any error in it. The readers have access to all the dataset and code to test it for themselves.

for centroid size – I just focus on the giraffomorpha – the graph shows very long red taxa, i.e. no change in more recent times – how can the evol. rate for centroid size then increase so dramatically in recent times in the next graph?

Red means high evolutionary rates, as shown on the evolutionary scale. The high rates span the last 10 million years or so on the giraffomorpha branches; there is no contradiction to the next graph where rates accelerate from then on.

: this may be completely out of line, but I would like to ask you to just check that the taxa are correctly assigned. The graph for the giraffids on the left is super-similar to the graph for the moschids on the right. the graph for the cervids on the left is very similar to that of the giraffids on the right. maybe these are coincidences, just make really sure it is not some bug. on the left, there are some details that seem to be identical for for the right, the super-increase in giraffids compared to the others is striking – is this real? or a data artefact?

We have checked, and confirm the right attribution of the taxa. You have access to all the data to verify it by yourself (full access to all the raw data, results and also the R script is given).

for a family that has had a distinct reduction in species numbers, it appears strange that the giraffids should have such a dramatic recent increase in evolutionary rates. I still do not know what to make of the difference in magnitude between the evol. rate measures in the two graphs.

We are not comparing the same things, one is morphological evolution and the other body mass evolution. It is normal that the magnitude changes. Indeed there is a huge increase for the giraffes at the end considering mass, but the largest giraffes on the evolutionary timescale are the most recent ones.

Figure 4 This graph does not correspond to any of the supplementary material

You can find a similar graph in Supplementary Material 1-4 ->RPanda PCA->
RPANDA_PCA_VS_temperature_Tragulina_pecoran_families.pdf

sincerely marcus clauss

Reviewer #3 (Remarks to the Author):

Thank you for adding the extra contextualisation to the discussion, it now reads much better. I am happy with the manuscript as it is now.

Thank you very much